# Invariant Deep Uplift Modeling for Incentive Assignment in Online Marketing via Probability of Necessity and Sufficiency

**Zexu Sun** [1]  **Qiyu Han** [2]  **Hao Yang** [1]  **Anpeng Wu** [3]  **Minqin Zhu** [3]  **Dugang Liu** [4]  **Chen Ma** [5]  **Yunpeng Weng** [6]  **Xing Tang** [7]  **Xiuqiang He** [7]

## Abstract

In online platforms, incentives (*e.g.*, discounts, coupons) are used to boost user engagement and revenue. Uplift modeling methods are developed to estimate user responses from observational data, often incorporating distribution balancing to address selection bias. However, these methods are limited by in-distribution testing data, which mirrors the training data distribution. In reality, user features change continuously due to time, geography, and other factors, especially on complex online marketing platforms. Thus, effective uplift modeling method for out-of-distribution data is crucial. To address this, we propose a novel uplift modeling method **I**nvariant **D**eep **U**plift **M**odeling, namely **IDUM**, which uses invariant learning to enhance out-of-distribution generalization by identifying causal factors that remain consistent across domains. IDUM further refines these features into necessary and sufficient factors and employs a masking component to reduce computational costs by selecting the most informative invariant features. A balancing discrepancy component is also introduced to mitigate selection bias in observational data. We conduct extensive experiments on public and real-world datasets to demonstrate IDUM's effectiveness in both in-distribution and out-of-distribution scenarios in online marketing. Furthermore, we also provide theoretical analysis and related proofs to support our IDUM's generalizability.

[1]Gaoling School of Artificial Intelligence, Renmin University of China [2]School of Statistics, Renmin University of China [3]Department of Computer Science and Technology, Zhejiang University [4]College of Computer Science and Software Engineering, Shenzhen University [5]Department of Computer Science, City University of Hong Kong [6]FiT, Tencent [7]School of Big data and Internet, Shenzhen Technology University. Correspondence to: Xing Tang <xing.tang@hotmail.com>, Xiuqiang He <hexiuqiang@sztu.edu.cn>.

*Proceedings of the 42$^{nd}$ International Conference on Machine Learning*, Vancouver, Canada. PMLR 267, 2025. Copyright 2025 by the author(s).

## 1. Introduction

With the development of online platforms, online marketing has become increasingly important and competitive (Liu et al., 2023; Sun et al., 2024a). Assigning appropriate incentives to users has become a pivotal strategy for enhancing user conversion rates and boosting revenue. These incentives generally contain carefully designed contents (*e.g.*, discounts, coupons). To achieve this purpose, uplift modeling has been proposed in recent years (Yao et al., 2021; Sun & Chen, 2024). For instance, Booking implements promotional strategies to enhance user satisfaction (Albert & Goldenberg, 2022), Meituan uses cash bonuses to stimulate user retention (Wang et al., 2023), Kuaishou utilizes virtual coins to increase users' playback duration (Ai et al., 2022). A primary challenge arises from the non-random assignment of incentives in observational data. For instance, platforms may assign incentives based on user age, indicating a tendency to target younger users to enhance profit margins, which is called selection bias.

In recent years, with the development of uplift modeling, there have been many works proposed to solve the above challenge (Zhang et al., 2021). These works can be divided into three research lines: 1) *Meta-learner-based*. The basic idea of this line is to estimate the users' responses by using existing predictive models as the base learner. Two of the most representative methods are S-Learner and T-Learner (Künzel et al., 2019), which adopt a global base learner and two base learners corresponding to the treatment and control groups, respectively. 2) *Tree (or Forest)-based*. The basic idea of this line is to employ a hierarchical tree structure to systematically partition the user population into sub-populations that exhibit sensitivity to specific treatments. An essential step involves modeling the uplift directly by applying diverse splitting criteria, including considerations of distribution divergences (Radcliffe & Surry, 2011) and expected responses (Zhao et al., 2017; Saito et al., 2020). 3) *Neural network-based*. The basic idea of this line is to leverage the power of neural networks to develop estimators that are both intricate and versatile in predicting the user's response. CFRNet (Shalit et al., 2017) uses the Integer Probabilistic Metric (IPM) to solve the selection bias

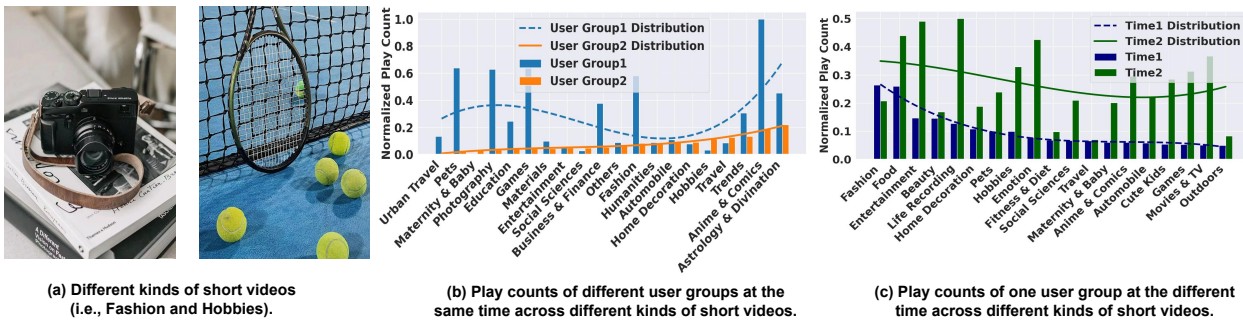

(a) Different kinds of short videos
(i.e., Fashion and Hobbies).

(b) Play counts of different user groups at the
same time across different kinds of short videos.

(c) Play counts of one user group at the different
time across different kinds of short videos.

Figure 1. The visualization of the out-of-distribution problem in the online short video platform.

in the latent space. Dragonnet (Shi et al., 2019) constructs the target regularizer for modeling the prediction biases of the user response, which can reduce the estimation errors of the uplift.

However, the main limitation of these methods is that they have only tested on data similar to the training data, which is known as the in-distribution (ID) data. In real-world applications, user features suffer continuous changes due to varying time and different geographical environments. Thus, concerns arise regarding the performance of these methods when applied to users whose feature distributions differ from those of the training data (Zhou et al., 2022). As shown in Figure 1, this problem is more severe in short video online marketing scenario, where the features of the users are more complex than others. In Figure 1(b), we can see a vast difference in the play counts between different user groups. The distribution of users' features may change over time, seasons, holidays, urban and rural areas, etc., resulting in the emergence of change in users' preferences, which also can be reflected in Figure 1(c). This can be regarded as the issue of out-of-distribution (OOD) data generalization. If we only use the previous methods on the specific data, we may get unreliable prediction results. Such unreliability may lead to inappropriate incentive assignments, posing a huge threat to users' experience. Therefore, an urgent demand is to develop uplift modeling methods that can effectively generalize to unseen testing data or different users.

In this paper, in order to solve the above challenges, we propose a novel uplift modeling method **I**nvariant **D**eep **U**plift **M**odeling, namely **IDUM**. Specifically, our IDUM proposes an invariant property learning part aimed at enhancing out-of-distribution uplift generalization by identifying causal features with invariant properties across domains. By separating these features into necessary and sufficient factors, we can learn the invariant features with finer granularity. Moreover, to reduce computational redundancy, we design a Gumbel-Softmax-based feature selection part to identify key subsets for invariant property learning. Additionally, to address the common selection bias issue of uplift esti-

mation over observational data, we introduce a balancing discrepancy part to balance the distributions between different treatment groups. Moreover, we conduct extensive experiments on various datasets to evaluate the effectiveness of our IDUM model.

## 2. Related Work

**Uplift Modeling.** Recent years have witnessed significant attention in uplift modeling (Sun et al., 2024b; 2025), particularly in the environment of online marketing (Zhang et al., 2021). This research predominantly focuses on model development and real-world applications. In cases of binary outcomes, the standard approach entails constructing two classification models (Künzel et al., 2019), which involve separate conditional probability models for treated and control users. This two-model strategy, compatible with any conventional estimator such as regression trees (Loh, 2011), is straightforward and versatile. However, it does not address the disparity in feature distributions between treatment and control groups. To overcome this, Künzel et al. (Künzel et al., 2019) introduced the X-Learner, which utilizes propensity scores (Caliendo & Kopeinig, 2008) to calculate a weighted average of two estimators, thus mitigating differences in feature distributions. For direct uplift modeling, a transformed response method (Athey & Imbens, 2016) was proposed, yet its success heavily relies on the accuracy of propensity scores. Regarding continuous outcomes, the Causal Forest (Davis & Heller, 2017), a random forest-like algorithm, employs causal trees (Darondeau & Degano, 1989) as its basic learner, offering a robust framework with theoretical support. With the emergence of deep learning in causal inference, many studies focusing on individual treatment effect (ITE) estimation have been introduced. TARNet (Shalit et al., 2017) features a two-head structure similar to the T-Learner, with a shared representation layer that facilitates the exchange of information between the heads. CFRNet employs distance metrics (MMD and WASS) based on TARNet's architecture to balance the representation across both heads. To address the

sample imbalance between treatment and control groups, Dragonnet (Shi et al., 2019) implements a tri-head structure, incorporating a separate head for learning the propensity score. Different from the above methods, our IDUM focuses on the distributional shift of the testing data in the uplift modeling.

**Invariant Learning.** In recent years, invariant learning has become a critical field of research in machine learning, focused on developing models that retain robust performance amidst distributional changes (Zhao et al., 2019). Many works on environmental generalization concentrate on identifying invariances across training environments (Zhou et al., 2022). Recently, a strategy that has made significant impacts involves learning features that enable a predictor to remain invariant across different environments (Creager et al., 2021; Lin et al., 2022; Hung & Chou, 2015), termed invariant learning in this paper. Such strategy is based on the theory of causality (Saengkyongam et al., 2023), where Structural Equation Models (Pearl, 2010) or causal graphs (Yao et al., 2021) are used to describe assumptions on the data generation process. The feature-conditioned label distribution invariance arises from the consistency of causal mechanisms across various environments. Recent theoretical studies on invariant learning examine its failure cases in environment generalization tasks where environments are known prior (Chen et al., 2022). Invariant risk minimization (IRM) methodologies (Rosenfeld et al., 2020) present an approach for acquiring knowledge of invariant variables and functions. Based on this, some other works (Krueger et al., 2021; Lu et al., 2021; Ahuja et al., 2021) further extend the IRM framework involves the incorporation of elements from game theory, variance penalization, information theory, as well as the integration of nonlinear prediction functions. Additionally, recent studies have applied the IRM framework to large neural networks, thereby contributing to its advancement (Rosenfeld et al., 2020; Gulrajani & Lopez-Paz, 2020). Based on the probability of necessity and sufficiency (PNS), some works design structures to discover the PNS features. CaSN employs probability of necessity and sufficiency (PNS) to extract environment-invariant information (Yang et al., 2024). Diverging from the aforementioned approaches, our IDUM focuses on the probability of necessity and sufficiency within generalized uplift modeling methods. Moreover, taking into account computational costs, we design additional structures to address this concern.

## 3. Preliminaries

### 3.1. Problem Setup

Following previous studies (Sun et al., 2023b; Zhu et al., 2023; He et al., 2024), we address the problem of uplift modeling within the framework of the Potential Outcome

Framework (Rubin, 2005). Our aim is to predict the uplift across multiple users. Generally, we consider the users used for training the model are sampled from the environment $e \in \mathcal{E}$, the target users are sampled from environment $e' \in \mathcal{E}$. With an observed dataset $\mathcal{D}^e = \{\boldsymbol{x}_i^e, t_i^e, y_i^e\}_{i=1}^N$, where $N$ represents the number of samples. For each user $i$, $\boldsymbol{x}_i^e \in \mathcal{X}^d$ denotes the features, $t_i^e \in \{0,1\}$ indicates the treatment assignment (1 for receiving an incentive, 0 otherwise), $y_i^e \in \mathcal{Y} \subset \mathbb{R}$ is the response variable. The target users are from a different environment $e' \in \mathcal{E}$, and $\{\boldsymbol{x}_i^{e'}\}_{i=1}^V$, where $V$ represents the number of samples. This dataset only considers the feature shift without the corresponding treatment assignment or responses. Our goal is to learn a model from the dataset $\mathcal{D}^e$ to predict the uplift for the dataset $\mathcal{D}^{e'}$ sampled from different environments $e' \in \mathcal{E}$. We treat this problem as the uplift prediction for *Out-of-Distribution* users. The objective of uplift modeling is to evaluate the effect of treatment $t_i^e$ on the response $y_i^e$ for a given user with features $\boldsymbol{x}_i$, specifically calculating the difference between treatment and control responses as follows:

$$\tau_i^e = y_i^e(1) - y_i^e(0), \tag{1}$$

In the real world, we can only observe one of the two responses for each user, that is, either $y_i^e(1)$ or $y_i^e(0)$ is the accessible factual response, the other is the counterfactual one remained unavailable.

For simplicity, subscript $i$ and superscript $e$ will be omitted where the context remains clear. Although the uplift or individual treatment effect in Eq. (1) cannot be directly identified due to the unobserved counterfactual, the conditional average treatment effect (CATE) offers a viable estimator for uplift under common assumptions (Shalit et al., 2017; Abrevaya et al., 2015). The CATE for a sub-population or individual is defined as:

$$\begin{aligned} \tau(\boldsymbol{x}) &= \mathbb{E}\left(Y(1) \mid \boldsymbol{X} = \boldsymbol{x}\right) - \mathbb{E}\left(Y(0) \mid \boldsymbol{X} = \boldsymbol{x}\right) \\ &= \mathbb{E}(Y \mid T = 1, \boldsymbol{X} = \boldsymbol{x}) - \mathbb{E}(Y \mid T = 0, \boldsymbol{X} = \boldsymbol{x})). \end{aligned} \tag{2}$$

After that, we can derive the uplift $\tau(x)$ by only leveraging the observational dataset.

### 3.2. Probability of Necessity and Sufficiency

Invariant learning is widely employed for out-of-distribution generalization (Arjovsky et al., 2019; Liu et al., 2021). The fundamental assumption of invariant learning posits that, given the features $\boldsymbol{X}$, response $Y$ and treatment $T$, only the environment invariant features $\boldsymbol{X}^c$ can reliably predict the response, while other environment-specific features $\boldsymbol{X}^v$ are spurious correlates of the response, thereby compromising the model's generalizability. Hence, invariant learning aims to identify domain-invariant features that satisfy the distributions of responses $P_e(Y|\boldsymbol{X}^c, T) = P_{e'}(Y|\boldsymbol{X}^c, T)$ across environments $e$ and $e'$.

To further analyze the invariant features, Probability of Necessity and Sufficiency (PNS) (Pearl, 2022) is proposed to describe the probability that event $A$ occurs if and only if event $B$ occurs, which separates the invariant features into necessary and sufficient features. This probability operates in two events to compute the probability that event $A$ is necessary and sufficient cause for event $B$. To better understand this, we have the following definitions:

**Definition 3.1** (Probability of necessity, PN)**.**

$$PN = P\left(B_{\bar{A}} \mid A, B\right), \qquad (3)$$

PN is the probability that, given that events $A$ and $B$ both occur initially, event $B$ does not occur (e.g., $\bar{B}$) after event $A$ is changed from occurring to not occurring (e.g., $\bar{A}$).

**Definition 3.2** (Probability of sufficiency, PS)**.**

$$PS = P\left(B_A \mid \bar{A}, \bar{B}\right), \qquad (4)$$

PS is the probability that, given that events $A$ and $B$ both did not occur initially, event $B$ occurs after event $A$ is changed from not occurring to occurring.

**Definition 3.3** (Probability of necessity and sufficiency, PNS)**.**

$$PNS = PN \cdot P(A, B) + PS \cdot P(\bar{A}, \bar{B}), \qquad (5)$$

PNS is the sum of PN and PS, each multiplied by the probability of its corresponding condition. PNS measures the probability that event $A$ is a necessary and sufficient cause for event $B$.

# 4. Methodology

In this section, we propose the IDUM model, a novel method to improve the out-of-distribution generalization for uplift modeling task. To begin with, we design an invariant property learning part to learn the sufficient and necessary features, capturing the invariant pattern behind data simultaneously. To reduce the computational cost, we employ a feature selection part with a neural network-based masking function. Finally, we introduce a balancing discrepancy part to address the selection bias issue in the incentive assignment process. The whole structure of our IDUM is shown in Figure 2. The notations used in this paper are presented in Appendix A.

## 4.1. Invariant Property Learning

As mentioned in Section 1, existing uplift modeling methods are hard to handle the distribution shift, therefore perform poorly in the out-of-distribution testing data. As proved in Zhou et al. (2021), the distribution shift issue is probably caused by the spurious correlation between the user features and the response. Therefore, an intuitive solution is to learn

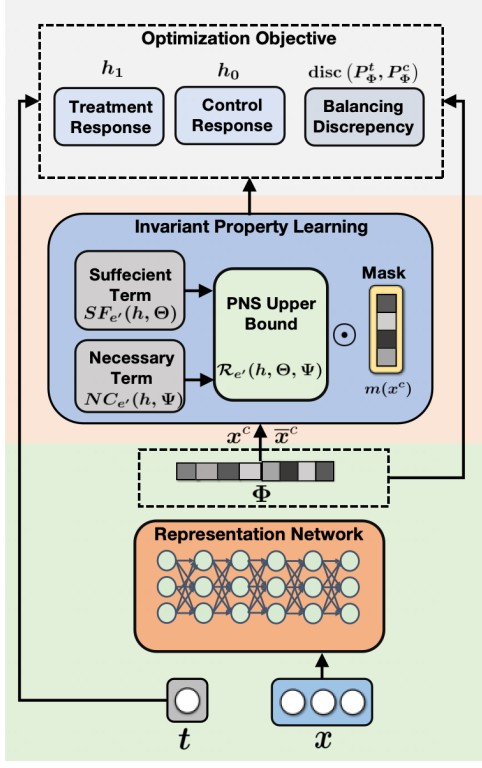

*Figure 2.* The whole structure of our IDUM, where $h_0$ and $h_1$ are the prediction heads of treatment and control groups. For simplicity, we use $h$ to represent the response prediction in Section 4.

the causal and invariant features to remove such spurious relationships. Technically, inspired by Pearl (2022); Yang et al. (2024), we design an invariant property learning part for enhancing the out-of-distribution uplift generalization. In detail, we propose the theories and the objective for extracting sufficient and necessary features based on PNS in the uplift generalization.

**PNS Risk and Upper Bound.** We begin by introducing the basic backbone $\Phi(x)$, which is constructed by a multi-layer perceptron (MLP) on the features of the treated and control samples. Based on the conceptions of sufficient and necessary features, we have the definition of PNS risk to learn the PNS value.

**Definition 4.1** (PNS Risk)**.** The PNS Risk is defined over the distribution of a target environment $e' \in \mathcal{E}$.

$$\mathcal{R}_{e'}(h, \Theta, \Psi) = \mathbb{E}_{(\boldsymbol{x}, t, y) \sim e'} \bigg[ \underbrace{\mathbb{E}_{\boldsymbol{x}^c \sim P_{e'}^\Theta(\boldsymbol{X}^c | \Phi(\boldsymbol{x}))} \mathbb{I}\left[h(\boldsymbol{x}^c, t) \neq y\right]}_{SF_{e'}(h, \Theta)} +$$

$$\underbrace{\mathbb{E}_{\overline{\boldsymbol{x}}^c \sim P_{e'}^\Psi(\boldsymbol{X}^c | \Phi(\boldsymbol{x}))} \mathbb{I}\left[h(\overline{\boldsymbol{x}}^c, t) = y\right]}_{NC_{e'}(h, \Psi)} \bigg],$$

$$(6)$$

where $SF_{e'}(h, \Theta)$ and $NC_{e'}(h, \Psi)$ denote the sufficient

and necessary terms, respectively. $\boldsymbol{x}^c$ and $\overline{\boldsymbol{x}}^c$ are the special implementations of $\boldsymbol{X}^c$, where $\boldsymbol{x}^c \neq \overline{\boldsymbol{x}}^c$. $\Theta$ and $\Psi$ are the parameters for learning the sufficient and necessary terms.

Variable $\boldsymbol{X}^c$ has a high probability of being the sufficient and necessary feature of $Y$ when the PNS value is small. However, computing the probability is a challenging problem for gathering counterfactual data (*i.e.*, $\boldsymbol{x}^c$ and $\overline{\boldsymbol{x}}^c$), which may be impractical or unattainable in real-world systems. Fortunately, the PNS defined on the counterfactual distribution can be directly estimated from the data under appropriate conditions (*i.e.*, Exogeneity and Monotonicity).

**Definition 4.2** (Exogeneity (Pearl, 2022)). $\boldsymbol{X}^c$ is exogenous relative to variable $Y$ w.r.t. source and target environments $e$ and $e'$, if the probability is identified by conditional probability $P_e\left(Y_{do(\boldsymbol{X}^c=\boldsymbol{x}^c)} = y \mid \boldsymbol{X}^c, T\right) = P_e(Y = y \mid \boldsymbol{X}^c = \boldsymbol{x}^c, T)$ and $P_{e'}\left(Y_{do(\boldsymbol{X}^c=\boldsymbol{x}^c)} = y \mid \boldsymbol{X}^c, T\right) = P_{e'}(Y = y \mid \boldsymbol{X}^c = \boldsymbol{x}^c, T)$.

**Definition 4.3** (Monotonicity (Pearl, 2022)). $Y$ is monotonic relative to $X$ if and only if either $P\left(Y_{do(\boldsymbol{X}^c=\boldsymbol{x}^c)} = y, Y_{do(\boldsymbol{X}^c=\overline{\boldsymbol{x}}^c)} \neq y \mid \boldsymbol{X}^c, T\right) = 0$ or $P\left(Y_{do(\boldsymbol{X}^c=\boldsymbol{x}^c)} \neq y, Y_{do(\boldsymbol{X}^c=\overline{\boldsymbol{x}}^c)} = y \mid \boldsymbol{X}^c, T\right) = 0$.

The definition of Exogeneity delineates that the disparity between the counterfactual data and conditional distributions diminishes when $\boldsymbol{X}^c$ is exogenous with respect to $Y$, while Monotonicity illustrates the monotonic effect of variable $\boldsymbol{X}^c$ on $Y$. Then, the identifiability of PNS can be described as the following lemma.

**Lemma 4.4.** *(Pearl, 2022) If $\boldsymbol{X}^c$ is exogenous relative to $Y$, and $Y$ is monotonic relative to $\boldsymbol{X}^c$, then*

$$
\begin{aligned}
PNS(\boldsymbol{x}^c, \overline{\boldsymbol{x}}^c) &= P_{e'}(Y = y \mid \boldsymbol{X}^c = \boldsymbol{x}^c, T) \\
&\quad - P_{e'}(Y = y \mid \boldsymbol{X}^c = \overline{\boldsymbol{x}}^c, T).
\end{aligned}
\tag{7}
$$

According to Lemma 4.4, the computation of PNS becomes feasible through observational data under the assumptions of Exogeneity and Monotonicity. This enables the quantification of PNS even in the absence of counterfactual data. The proof of Lemma 4.4 is provided by Pearl (2009), and its applicability to our problem setting can be directly extended. Subsequently, we seamlessly incorporate the measure of Monotonicity into the PNS risk by deriving an upper bound.

**Proposition 4.5** (Upper Bound of PNS Risk). *Given the target environment, let the Monotonicity measurement be:*

$$
\begin{aligned}
M_{e'}^h(\Theta, \Psi) = \mathbb{E}_{(\boldsymbol{x},t,y)\sim e'} \mathbb{E}_{\boldsymbol{x}^c \sim P_{e'}^\Theta(\boldsymbol{X}^c|\Phi(\boldsymbol{x}))} \mathbb{E}_{\overline{\boldsymbol{x}}^c \sim P_{e'}^\Psi(\boldsymbol{X}^c|\Phi(\boldsymbol{x}))} \\
\mathbb{I}\left[h(\boldsymbol{x}^c, t) \neq h(\overline{\boldsymbol{x}}^c, t)\right],
\end{aligned}
$$

*Then, we can get:*

$$
\begin{aligned}
R_{e'}(h, \Theta, \Psi) &= M_{e'}^h(\Theta, \Psi) + 2SF_{e'}(h, \Theta)NC_{e'}(h, \Psi) \\
&\leq M_{e'}^h(\Theta, \Psi) + 2SF_{e'}(h, \Theta).
\end{aligned}
\tag{8}
$$

We present the proof in Appendix B. The upper bound for the PNS risk in Eq. (8) comprises two components: (1) the sufficiency evaluator $SF_{e'}(h, \Theta)$ and (2) the monotonicity measurement $M_{e'}^h(\Theta, \Psi)$. In this upper bound, the necessary term $NC_{e'}(h, \Psi)$ is assimilated into the Monotonicity measurement $M_{e'}^h(\Theta, \Psi)$. The minimization process of Eq. (6) with respect to its upper bound, Eq. (8), entails ensuring the satisfaction of Monotonicity.

**OOD Generalization.** In out-of-distribution (OOD) uplift generalization, only source data gathered from environment $e$ are available, whereas the target environment $e'$ remains inaccessible throughout the optimization procedure. Consequently, directly assessing the risk on the target environment, denoted as $R_{e'}(h, \Theta, \Psi)$, is unfeasible. Hence, we establish the linkage between the risks associated with the source environment, $R_e(h, \Theta, \Psi)$, and the target environment, $R_{e'}(h, \Theta, \Psi)$.

We introduce the divergence measurement known as $\beta$ divergence (Ganin et al., 2016) and apply variational approximation to weight the term $R_{e'}(h, \Theta, \Psi)$. Formally, $\beta$ divergence quantifies the dissimilarity between environments $e$ and $e'$, as formally defined below.

$$
\beta_q(e'\|e) = \left[ \mathop{\mathbb{E}}_{(\boldsymbol{x},t,y)\sim e} \left( \frac{e'(\boldsymbol{x}, t, y)}{e(\boldsymbol{x}, t, y)} \right)^q \right]^{\frac{1}{q}},
\tag{9}
$$

Based on $\beta_q(e'\|e)$, we connect the risks on the source and target environments by Theorem 4.6.

**Theorem 4.6.** *The risk in the target environment is bounded by the risk in the source environment:*

$$
\begin{aligned}
R_{e'}(h, \Theta, \Psi) &\leq \lim_{q\to+\infty} \beta_q(e'\|e) \left( \left[ M_{e'}^h(\Theta, \Psi) \right]^{1-\frac{1}{q}} \right. \\
&\quad \left. + 2 \left[ SF_{e'}(h, \Theta) \right]^{1-\frac{1}{q}} \right) + \xi_{e'\backslash e}(\boldsymbol{X}, T, Y),
\end{aligned}
$$

*where $\xi_{e'\backslash e}(\boldsymbol{X}, T, Y) := P_{e'}(\boldsymbol{X} \times T \times Y \notin \mathrm{supp}(e)) \cdot \sup(R_{e'\backslash e}(h, \Theta, \Psi))$.*

We provide the proof in Appendix B. Here $\mathrm{supp}(e)$ is the support set of source environment distribution $P_e(\boldsymbol{X})$,

$$
\begin{aligned}
R_{e'\backslash e}(h, \Theta, \Psi) = \mathbb{E}_{(\boldsymbol{x},t,y)\sim P_{e'}(\boldsymbol{X}\times T\times Y\notin\mathrm{supp}(e))} [SF_{e'}(h, \Theta) \\
+ NC_{e'}(h, \Psi)].
\end{aligned}
$$

Theorem 4.6 establishes a connection between the risks of the source and target environments. Within Theorem 4.6, $\xi_{e'\backslash e}(\boldsymbol{X}, T, Y)$ delineates the expectation of the worst risk in unknown area, where the data sample $(\boldsymbol{x}, t, y)$ does not belong to the support set $\mathrm{supp}(e)$ of the source environment. When the observations $\boldsymbol{X}$ in $e$ and $e'$ share the same support set, the term $\xi_{e'\backslash e}(\boldsymbol{X}, T, Y)$ tends towards 0.

In many real-world uplift modeling scenarios where the distribution $e$ is not directly accessible, we examine the

association between the expected risk based on the source environment distribution and the empirical risk computed from the data of the source environment. Similarly, we can assess the empirical risks with respect to $\widetilde{SF}_e(h, \Theta)$ and $\widetilde{NC}_e(h, \Psi)$. Subsequently, we leverage PAC-learning (Denis, 1998) to formulate a theorem in the following, establishing an upper bound for the discrepancy between empirical risk and expected risk.

**Theorem 4.7.** *Given parameters* $\Theta, \Psi$, *for any* $h$, *prior distribution* $\pi_{\boldsymbol{X}^c} = P_e(\boldsymbol{X}^c)$ *and* $\pi_{\overline{\boldsymbol{X}}^c} = P_e(\overline{\boldsymbol{X}}^c)$ *which make* $\mathbb{E}_{e^n} \mathrm{KL}\left(P_e^\Theta(\boldsymbol{X}^c \mid \Phi(\boldsymbol{X}) = \Phi(\boldsymbol{x})) \| \pi_{\boldsymbol{X}^c}\right)$ *and* $\mathbb{E}_{e^n} \mathrm{KL}\left(P_e^\Psi(\overline{\boldsymbol{X}}^c \mid \Phi(\boldsymbol{X}) = \Phi(\boldsymbol{x})) \| \pi_{\overline{\boldsymbol{X}}^c}\right)$ *both lower than a positive constant* $C$, *then with a probability at least* $1 - \epsilon$ *over source environment data,*

*(1)* $\left| SF_e(h, \Theta) - \widetilde{SF}_e(h, \Theta) \right|$ *is upper bounded by*

$$\mathbb{E}_{e^n} \mathrm{KL}\left(\tilde{P}_e^\Theta(\boldsymbol{X}^c \mid \Phi(\boldsymbol{X}) = \Phi(\boldsymbol{x})) \| \pi_{\boldsymbol{X}^c}\right) + \frac{\ln(n/\epsilon)}{2(n-1)} + C,$$

*where* $n$ *is the number of the source data.*

*(2)* $\left| M_e^h(\Theta, \Psi) - \widetilde{M}_e^h(\Theta, \Psi) \right|$ *is upper bounded by*

$$\mathbb{E}_{e^n} \mathrm{KL}\left(\tilde{P}_e^\Theta(\boldsymbol{X}^c \mid \Phi(\boldsymbol{X}) = \Phi(\boldsymbol{x}))\right)$$
$$+ \mathbb{E}_{e^n} \mathrm{KL}\left(\tilde{P}_e^\Psi(\overline{\boldsymbol{X}}^c \mid \Phi(\boldsymbol{X}) = \Phi(\boldsymbol{x})) \| \pi_{\overline{\boldsymbol{X}}^c}\right) + \frac{\ln(n/\epsilon)}{2(n-1)} + 2C.$$

The proof is provided in Appendix B. Theorem 4.7 illustrates that as the sample size increases and the terms involving KL divergence diminish, the empirical risk computed on the source environment dataset converges towards the expected risk. By combining Theorems 4.6 and 4.7, we can get the expected PNS risk on the target distribution by leveraging the empirical risk computed on the source data.

### 4.2. Feature Selection

After we have designed the above PNS risk upper bound, a problem that needs to be solved is that the dimension of the shared feature representation $\Phi(\boldsymbol{x})$ in the uplift model is always large, thus, the computational cost to find $\boldsymbol{x}^c$ is big. To address this challenge, we employ a neural network-based masking function, denoted as $\hat{w}(\cdot)$, which determines the contribution of each feature in the uplift estimation. Additionally, we utilize the Gumbel-Softmax trick (Jang et al., 2016) to constrain the model, enabling the acquisition of an approximate $k$-hot mask vector $m(\boldsymbol{x}^c)$. The $\kappa$-ratio features with the greatest contribution are identified as the desired key features, while the remaining features are deemed irrelevant or redundant. The formulation of $m(\boldsymbol{x}^c)$ is as follows:

$$m(\boldsymbol{x}^c) = \text{Gumbel-Softmax}(\hat{w}(\boldsymbol{x}^c), \kappa H) \in \mathbb{R}^H, \quad (10)$$

where $k = \lfloor \kappa H \rfloor \in \mathbb{Z}_+$ is the number of features expected to be obtained, $H$ is the number of input features. Specifically, in Eq. (10), let $z = \hat{w}(\boldsymbol{x}^c) \in \mathbb{R}^H$ represent a probability vector. For any feature dimension $j \in 1, \ldots, H$, it holds that $z_j \geq 0$ and $\sum_j z_j = 1$. With a predefined temperature $\zeta > 0$, the computation of each feature dimension in the mask vector can be expressed below (Lv et al., 2022).

$$m_j = \max_{l \in \{1, \ldots, k\}} \frac{\exp\left(\left(\log z_j + \eta_j^l\right)/\zeta\right)}{\sum_{j'=1}^N \exp\left(\left(\log z_{j'} + \eta_{j'}^l\right)/\zeta\right)}, \quad (11)$$

where $l \in \{1, \ldots, k\}$ denotes the index of the selected feature, $\eta_j^l = -\log\left(-\log u_j^l\right)$, and $u_j^l \sim \text{Uniform}(0, 1)$ denotes a uniformly distributed sampling. Finally, we can get the masked features $\boldsymbol{x}_m^c$ by multiplying the original features $\boldsymbol{x}^c$ with the resulting mask vector $m(x)$.

$$\boldsymbol{x}_m^c = \boldsymbol{x}^c \odot m(\boldsymbol{x}^c), \quad (12)$$

where $\odot$ denotes the element-wise multiplication. Remarkably, the same procedure is applied to $\overline{\boldsymbol{x}}^c$.

### 4.3. Balancing Discrepancy

Up to now, we have adopted invariant learning to enhance out-of-distribution uplift generalization. However, there remains the issue of selection bias, which is common when estimating uplift on observational datasets. To address this, we apply the distribution discrepancy regularizer from CFR-Net (Shalit et al., 2017), which achieves a balanced representation by minimizing the distributional differences between treatment and control groups. A typical metric used for measuring the distribution discrepancy is the Integral Probability Metric (IPM) (Müller, 1997; Sriperumbudur et al., 2012), which is formally defined as,

$$\text{disc}\left(P_\Phi^t, P_\Phi^c\right) = \sup_{h_0, h_1 \in \mathcal{H}} \left| \mathbb{E}_{\boldsymbol{x} \sim P_\Phi^t}[h_1(\boldsymbol{x}, t)] - \mathbb{E}_{\boldsymbol{x} \sim P_\Phi^c}[h_0(\boldsymbol{x}, t)] \right|, \quad (13)$$

where $P_\Phi^t = \{\Phi(\boldsymbol{x})\}_{t=1}$ and $P_\Phi^c = \{\Phi(\boldsymbol{x})\}_{t=0}$ denote the feature distributions of the treatment and control groups in the representation space, respectively. $h_0(\boldsymbol{x}, t)$ and $h_1(\boldsymbol{x}, t)$ represent the prediction heads, which are optimized by cross entropy loss. For ease of understanding, we use $h(\boldsymbol{x}, t)$ to represent them in the following description. For function families $\mathcal{H}$ with sufficient richness, when the discrepancy between the distributions $P_\Phi^t$ and $P_\Phi^c$ equals zero, it follows that $P_\Phi^t = P_\Phi^c$ holds (Shalit et al., 2017). Consequently, minimizing this discrepancy entails aligning the means in the feature space.

### 4.4. Optimization and Training Procedure

From above description, we know that the learning process of $P^\Psi(\overline{\boldsymbol{X}}^c \mid \Phi(\boldsymbol{X}) = \Phi(\boldsymbol{x}))$, coupled with the feature selection outlined in Section 4.2, can solve the uplift generalization problem. Throughout the learning phase, our

*Table 1.* Overall comparison between our IDUM and the baselines on ID Lazada and Production datasets. We report the results over five random seeds. The best results and second best results are **bold** and underlined, respectively.

| Method | Lazada Dataset (ID) | | | Production Dataset(ID) | | |
|---|---|---|---|---|---|---|
| | AUUC | QINI | KENDALL | AUUC | QINI | KENDALL |
| S-Learner | $0.0117 \pm 0.0029$ | $0.3826 \pm 0.0377$ | $0.1073 \pm 0.0141$ | $0.2450 \pm 0.0232$ | $0.3292 \pm 0.0258$ | $0.1643 \pm 0.0169$ |
| T-Learner | $0.0125 \pm 0.0027$ | $0.4442 \pm 0.0245$ | $0.1197 \pm 0.0140$ | $0.2461 \pm 0.0229$ | $0.3486 \pm 0.0247$ | $0.1671 \pm 0.0161$ |
| TARNet | $0.0171 \pm 0.0026$ | $0.4457 \pm 0.0217$ | $0.1372 \pm 0.0121$ | $0.2493 \pm 0.0218$ | $0.3428 \pm 0.0227$ | $0.1698 \pm 0.0173$ |
| CFRNet-mmd | $0.0271 \pm 0.0036$ | $0.4610 \pm 0.0219$ | $0.1698 \pm 0.0132$ | $0.2691 \pm 0.0224$ | $0.3408 \pm 0.0246$ | $0.1915 \pm 0.0139$ |
| CFRNet-wass | **$0.0273 \pm 0.0034$** | $0.4465 \pm 0.0209$ | $0.1753 \pm 0.0137$ | $0.2579 \pm 0.0223$ | $0.3503 \pm 0.0246$ | $0.1921 \pm 0.0147$ |
| DragonNet | $0.0241 \pm 0.0037$ | $0.4392 \pm 0.0292$ | $0.1444 \pm 0.0132$ | $0.2527 \pm 0.0201$ | $0.3476 \pm 0.0230$ | $0.1747 \pm 0.0173$ |
| EUEN | $0.0267 \pm 0.0033$ | $0.4456 \pm 0.0237$ | $0.1523 \pm 0.0113$ | $0.2793 \pm 0.0209$ | $0.3693 \pm 0.0234$ | $0.2004 \pm 0.0193$ |
| UniTE | $0.0189 \pm 0.0030$ | $0.4563 \pm 0.0283$ | $0.1457 \pm 0.0144$ | $0.2408 \pm 0.0211$ | $0.3681 \pm 0.0225$ | $0.1997 \pm 0.0167$ |
| TEED | $0.0120 \pm 0.0026$ | $0.4395 \pm 0.0234$ | $0.1256 \pm 0.0129$ | $0.2492 \pm 0.0239$ | $0.3515 \pm 0.0241$ | $0.1695 \pm 0.0167$ |
| **IDUM** | $0.0270 \pm 0.0023$ | **$0.4646 \pm 0.0184$** | **$0.1798 \pm 0.0137$** | **$0.2907 \pm 0.0248$** | **$0.3700 \pm 0.0255$** | **$0.2091 \pm 0.0154$** |

objective is to minimize the risk under the worst-case scenario induced by $\overline{X}_m^c$, namely, the maximum PNS risk resulting from the selection of $P^\Psi(\overline{X}_m^c \mid \Phi(X) = \Phi(x))$. Minimizing the upper bounds established in Theorem 4.6 and 4.7 can be simulated through the following optimization procedure:

$$\min_{\Theta, h, m, \Phi} \max_{\Psi} \widetilde{M}_e^{h,m}(\Theta, \Psi) + \widetilde{SF}_e(h, m, \Theta)$$
$$+ \alpha \operatorname{disc}\left(P_\Phi^t, P_\Phi^c\right) + \lambda L_{\mathrm{KL}}, \quad (14)$$
$$\text{subject to} \quad \|x_m^c - \overline{x}_m^c\|_2 > \delta,$$

where $L_{\mathrm{KL}} = \mathbb{E}_{e^n} \mathrm{KL}\left(\tilde{P}_e^\Theta\left(X_m^c \mid \Phi(X) = \Phi(x)\|\pi_{X_m^c}\right)\right)$ $+ \mathbb{E}_{e^n} \mathrm{KL}\left(\tilde{P}_e^\Psi\left(\overline{X}_m^c \mid \Phi(X) = \Phi(x)\|\pi_{\overline{X}_m^c}\right)\right)$. $h(x_m^c, t)$ is the prediction head of the response variable, and the optimization function can be the cross entropy loss. The constraint $\|x_m^c - \overline{x}_m^c\|_2 > \delta$ is established to ensure $\delta$-Semantic Separability (Yang et al., 2024). This criterion stipulates that the semantic meaning should be discernible between $x_m^c$ and $\overline{x}_m^c$ when their distance exceeds a certain threshold. The absence of this constraint may result in nearly identical values representing vastly different semantic information, leading to inherently unstable and chaotic data. Consequently, the final optimization objective is derived as follows:

$$\mathcal{L}_{\mathrm{IDUM}} = \min_{\Theta, h, m, \Phi} \max_{\Psi} \widetilde{M}_e^{h,m}(\Theta, \Psi) + \widetilde{SF}_e(h, m, \Theta)$$
$$+ \alpha \operatorname{disc}\left(P_\Phi^t, P_\Phi^c\right) + \beta \|x_m^c - \overline{x}_m^c\|_2 + \lambda L_{\mathrm{KL}}. \quad (15)$$

## 5. Experiments

### 5.1. Experimental Setups

**Datasets.** *Lazada dataset* (Zhong et al., 2022). We utilize a large-scale production dataset obtained from real voucher distribution operations at Lazada, a prominent e-commerce platform in Southeast Asia (SEA) operated by

the Alibaba Group. In the actual production setting, treatment assignment is selective owing to operational targeting strategies. Our training dataset comprises data characterized by significant treatment bias collected under these conditions. Additionally, we have a slightly smaller subset of users unaffected by the targeting strategies, where treatment assignment follows randomized controlled trials (RCT) for out-of-distribution (OOD) testing.

*Production dataset.* This dataset comes from an industrial production environment, specifically one of the largest short-video platforms in China. Clarity serves as a crucial user experience metric for such platforms. A decrease in clarity can significantly impact user satisfaction on the platform. Hence, we conducted random experiments over the course of a week, wherein high-clarity videos ($t = 1$) were provided to the treatment group, while low-clarity videos ($t = 0$) were provided to the control group. We collected the total play counts of users' short videos over the week and quantified the impact of resolution degradation on user experience. To enhance data robustness and mitigate response variance, we discretized each user's response (i.e., total play counts) into four levels. Furthermore, for each user group, we collect data at two distinct time points to facilitate out-of-distribution (OOD) testing. Moreover, the dataset statistics and visualization are presented in Appendix A.

**Baselines and Metrics** In this paper, we compare our method with the commonly used uplift modeling methods, **S-Learner** (Künzel et al., 2019), **T-Learner** (Künzel et al., 2019), **TARNet** (Shalit et al., 2017), **CFRNet** (Shalit et al., 2017), **DragonNet** (Shi et al., 2019), **EUEN** (Ke et al., 2021), **UniTE** (Liu & Hou, 2023) and **TEED** (Sun et al., 2023a). Following the previous works (Sun et al., 2023b; 2024c), we evaluate the performance of the methods by AUUC, QINI and KENDALL. More details are presented in Appendix A.

**Implementation Details** We implement all baselines and our IDUM based on Pytorch 1.10, with Adam as the opti-

*Table 2.* Overall comparison between our IDUM and the baselines on OOD Lazada and Production datasets. We report the results over five random seeds. The best results and second best results are **bold** and underlined, respectively.

| Method | Lazada Dataset (OOD) | | | Production Dataset (OOD) | | |
|---|---|---|---|---|---|---|
| | AUUC | QINI | KENDALL | AUUC | QINI | KENDALL |
| S-Learner | $0.0093 \pm 0.0028$ | $0.3581 \pm 0.0393$ | $0.1036 \pm 0.0137$ | $0.2444 \pm 0.0229$ | $0.3284 \pm 0.0263$ | $0.1632 \pm 0.0172$ |
| T-Learner | $0.0101 \pm 0.0027$ | $0.4392 \pm 0.0228$ | $0.1032 \pm 0.0124$ | $0.2442 \pm 0.0235$ | $0.3411 \pm 0.0244$ | $0.1617 \pm 0.0159$ |
| TARNet | $0.0104 \pm 0.0025$ | $0.4448 \pm 0.0204$ | $0.1081 \pm 0.0118$ | $0.2365 \pm 0.0205$ | $0.3347 \pm 0.0231$ | $0.1684 \pm 0.0161$ |
| CFRNet-mmd | $0.0262 \pm 0.0037$ | $0.4618 \pm 0.0210$ | $0.1737 \pm 0.0126$ | $0.2665 \pm 0.0214$ | $0.3383 \pm 0.0228$ | $0.1894 \pm 0.0143$ |
| CFRNet-wass | $0.0269 \pm 0.0036$ | $0.4332 \pm 0.0219$ | $0.1742 \pm 0.0137$ | $0.2583 \pm 0.0206$ | $0.3391 \pm 0.0237$ | $0.1912 \pm 0.0157$ |
| DragonNet | $0.0224 \pm 0.0035$ | $0.4179 \pm 0.0279$ | $0.1040 \pm 0.0124$ | $0.2423 \pm 0.0214$ | $0.3287 \pm 0.0212$ | $0.1724 \pm 0.0169$ |
| EUEN | $0.0250 \pm 0.0032$ | $0.4270 \pm 0.0225$ | $0.1428 \pm 0.0107$ | $0.2755 \pm 0.0210$ | $0.3632 \pm 0.0220$ | $0.1995 \pm 0.0180$ |
| UniTE | $0.0122 \pm 0.0028$ | $0.4499 \pm 0.0291$ | $0.1284 \pm 0.0131$ | $0.2298 \pm 0.0221$ | $0.3568 \pm 0.0215$ | $0.1899 \pm 0.0154$ |
| TEED | $0.0102 \pm 0.0031$ | $0.4239 \pm 0.0268$ | $0.1089 \pm 0.0128$ | $0.2237 \pm 0.0297$ | $0.3275 \pm 0.0223$ | $0.1682 \pm 0.0153$ |
| **IDUM** | **$0.0274 \pm 0.0026$** | **$0.4633 \pm 0.0191$** | **$0.1790 \pm 0.0152$** | **$0.2847 \pm 0.0265$** | **$0.3681 \pm 0.0245$** | **$0.2051 \pm 0.0166$** |

mizer and a maximum iteration count of 50. We use the QINI as a reference to search for the best hyper-parameters for all baselines and our model. We also adopt an early stopping mechanism with a patience of 5 to avoid over-fitting to the training set. Furthermore, we utilize the hyper-parameter search library Optuna (Akiba et al., 2019) to accelerate the tuning process, all experiments are implemented on NVIDIA A40 and Intel(R) Xeon(R) 5318Y Gold CPU @ 2.10GHz.

### 5.2. Overall Performance

To evaluate the performance of our IDUM on both in-distribution (ID) and out-of-distribution (OOD) testing data, we conduct several experiments on the Lazada dataset and the Production dataset. The results are reported in Table 1 and 2, where we can observe: 1) Among all the baseline methods, the S-learner, T-learner, and TARNet exhibit the poorest performance across both ID and OOD experiments. We speculate that their simplistic model architectures, lacking further constraints, struggle to mitigate both the selection bias and the distribution shift between training and testing sets, consequently leading to inaccurate uplift prediction in real-time marketing scenarios. 2) CFRNet-wass and CFR-mmd consistently demonstrate superior performance compared to other baselines. This can be attributed to their utilization of Integral Probability Metrics, such as Maximum Mean Discrepancy and Wasserstein distance, which facilitate the mitigation of selection bias between treatment and control groups. However, they still exhibit challenges in coping with the distribution shift problem, as evidenced by their comparatively lower performance in OOD experiments. 3) Encouragingly, our IDUM consistently outperforms all baselines across both the Lazada dataset and the Production dataset, particularly on OOD testing data. This demonstrates the effectiveness of the carefully designed invariant property learning part within IDUM, which encompasses the PNS risk and adversarial feature selection components, in robustifying the uplift model in a collaborative manner.

Therefore, our proposed IDUM can be considered as an effective generalized uplift model in online marketing.

### 5.3. Ablation Study

In this section, we conduct ablation studies on the OOD testing data to assess the necessity of each component in our proposed IDUM model. Specifically, we sequentially remove the Integral Probability Metrics-based representation Balancing Discrepancy component (BD), the Invariant Property Learning with PNS risk (IPL), and the Feature Selection with Gumbel-Softmax of IPL (IPL-FS). We then tailor three variants of IDUM, namely w/o BD, w/o IPL, and w/o IPL-FS. From the results reported in Table 3, it is evident that removing any part of IDUM leads to performance degradation. Firstly, the IPM regularization for representation balancing helps address the internal selection bias between the treatment group and the control group. Secondly, the IPL part, serving as a crucial component for managing the distribution shift between training and testing data, significantly contributes to the accurate estimation of uplift, as evidenced by the substantial performance drop when it is removed from IDUM. Lastly, the adversarial feature selection part aids in efficiently learning the invariant property hidden in the data, thereby further enhancing the robustness of our IDUM model in out-of-distribution scenarios.

### 5.4. Sensitivity Analysis

In this section, we conduct out-of-distribution (OOD) experiments to analyze the sensitivity of each important hyperparameter in our framework. Specifically, we vary the weight of IPM with $\alpha \in \{0.001, 0.005, 0.01, 0.05, 0.1\}$, the weight of the semantic constraint with $\beta \in \{0.001, 0.005, 0.01, 0.05, 0.1\}$, the weight of KL divergence with $\lambda \in \{0.01, 0.05, 0.1, 0.2, 0.3\}$, and the value of temperature with $\zeta \in \{0.5, 1, 1.5, 2, 2.5\}$. The results are presented in Figure 3. It is evident that extreme values of $\alpha$ for representation balancing regularization may adversely affect the model performance. Regarding the se-

*Table 3.* Ablation study of our IDUM. We report the results over five random seeds. The best results and second best results are **bold** and underlined, respectively.

| Method | Production Dataset (ID) | | | Production Dataset (OOD) | | |
|---|---|---|---|---|---|---|
| | AUUC | QINI | KENDALL | AUUC | QINI | KENDALL |
| **IDUM** | **0.2907** ± 0.0248 | **0.3700** ± 0.0255 | **0.2091** ± 0.0154 | **0.2847** ± 0.0265 | **0.3681** ± 0.0245 | **0.2051** ± 0.0166 |
| w/o BD | 0.2863 ± 0.0246 | 0.3658 ± 0.0217 | 0.2041 ± 0.0277 | 0.2710 ± 0.0227 | 0.3383 ± 0.0264 | 0.1801 ± 0.0153 |
| w/o IPL | 0.2680 ± 0.0206 | 0.3531 ± 0.0236 | 0.1939 ± 0.0162 | 0.2732 ± 0.0235 | 0.3475 ± 0.0204 | 0.1781 ± 0.0173 |
| w/o IPL-FS | 0.2891 ± 0.0224 | 0.3651 ± 0.0214 | 0.2087 ± 0.0159 | 0.2737 ± 0.0251 | 0.3242 ± 0.0253 | 0.2039 ± 0.0124 |

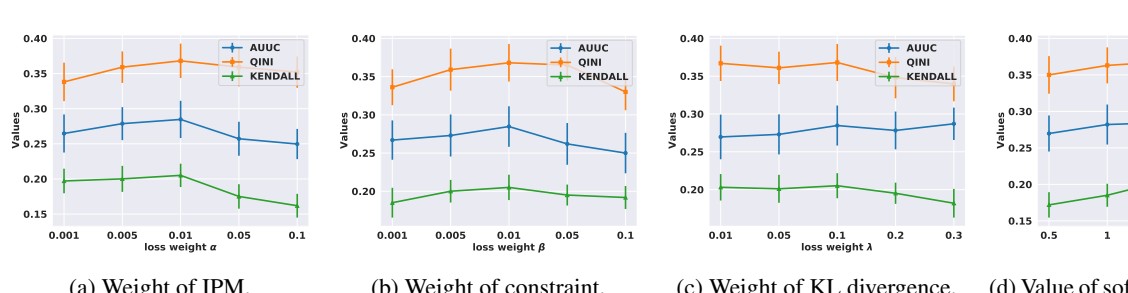

| (a) Weight of IPM. | (b) Weight of constraint. | (c) Weight of KL divergence. | (d) Value of softmax temperature. |

*Figure 3.* Sensitivity analysis of our IDUM on the OOD Production dataset. We report the results over five random seeds.

mantic constraint, the best uplift estimation performances are consistently achieved with a moderate value of $\beta$. Additionally, our IDUM model demonstrates varying tendencies across these three metrics when the value of $\lambda$ for KL divergence in invariant property learning is adjusted, indicating the need for careful tuning in real-world scenarios. Lastly, the IDUM model exhibits robustness to fluctuations in the softmax temperature $\zeta$, consistently selecting crucial features effectively.

## 6. Conclusion

In this paper, we propose a novel method called **I**nvariant **D**eep **U**plift **M**odeling, namely IDUM, to construct a generalized uplift model for incentive assignment in online marketing. Our proposed IDUM incorporates three key components to enhance out-of-distribution uplift generalization. Firstly, an invariant property learning part is designed to identify necessary and sufficient features with domain-invariant characteristics. Secondly, to optimize computational efficiency, we implement a Gumbel-Softmax-based feature selection mechanism that identifies crucial feature subsets for invariant learning. Thirdly, a balancing discrepancy component is introduced to mitigate distributional differences across treatment groups. Through extensive empirical evaluations and theoretical analysis, we demonstrate the effectiveness of our IDUM and provide rigorous generalization guarantees for it.

## Acknowledgments

We thank the support of SZTU University Research Project (No. 20251061020002).

## Impact Statement

The proposed IDUM to enhance uplift modeling for incentive allocation in online marketing by addressing distribution shifts and selection bias, which holds significant societal implications. Ethically, IDUM's ability to generalize across diverse user populations could promote fairer incentive distribution by reducing biases tied to spurious correlations (e.g., age or geography). This aligns with ethical AI principles by ensuring equitable access to promotional benefits. However, reliance on user data necessitates stringent privacy safeguards to prevent misuse of sensitive information. Future deployment should incorporate transparency mechanisms to audit algorithmic decisions and mitigate unintended discrimination.

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

*Table 4.* Statistics of the public dataset Lazada and real-world dataset Production.

| Dataset | Features | Training Data | | | Testing Data (ID) | | | Testing Data (OOD) | | |
|---|---|---|---|---|---|---|---|---|---|---|
| | | Treated | Control | Total | Treated | Control | Total | Treated | Control | Total |
| Lazada | 83 | 0.64 M | 2.27 M | 4.17 M | 0.28M | 0.32M | 0.60M | 0.47M | 0.43 M | 0.91 M |
| Production | 104 | 1.91M | 4.22M | 6.13M | 0.56M | 0.61M | 1.17M | 0.37M | 0.40M | 0.77M |

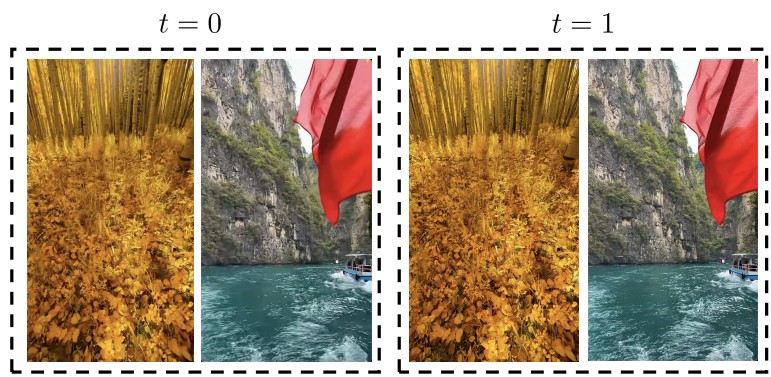

*Figure 4.* The visualization of the Production dataset. As $t$ increases, the clarity of the video correspondingly enhances.

## A. Additional Experiments

### A.1. Dataset statistics

In this section, we provide the statistics of the datasets that we used in our work, where the selection bias and out-of-distribution issues exist. The statistics of those two datasets are summarized in Table 4. Moreover, the visualization of the Production dataset is provided in Figure 4.

### A.2. Baselines and Metrics

We present a detailed description of the baselines we used in the following:

- **S-Learner** (Künzel et al., 2019): S-Learner is a kind of meta-learner method that uses a single estimator to estimate the response without giving the treatment a particular role.

- **T-Learner** (Künzel et al., 2019): T-Learner is similar to S-Learner, which uses two estimators for the treatment and control groups, respectively.

- **TARNet** (Shalit et al., 2017): TARNet is a commonly used neural network-based uplift model. It uses the shared bottom network to extract feature information.

- **CFRNet** (Shalit et al., 2017): CFRNet (*i.e.*, CFRNet-wass, CFRNet-mmd) applies an additional loss to TARNet, which forces the learned treated and control feature distributions to be closer.

- **DragonNet** (Shi et al., 2019): DragonNet exploits the sufficiency of the propensity score for estimation adjustment and uses a regularization procedure based on the non-parametric estimation theory.

- **EUEN** (Ke et al., 2021): EUEN is an explicit uplift modeling approach, which can correct the exposure bias.

- **UniTE** (Liu & Hou, 2023): UniTE adopts the Robinson Decomposition (Bratteli & Robinson, 2012) framework, and design a MMoE (Ma et al., 2018) based structure for uplift prediction.

- **TEED** (Sun et al., 2023a): TEED is a novel direct learning framework along with distribution adaptation and reliable scoring modules, which can solve the treatment effect estimation problem across environments.

*Table 5.* Notation Table of the notations in this work.

| Notation | Definition |
|---|---|
| $e$ and $e'$ | Two different environments |
| $\boldsymbol{X}$ | User features |
| $\boldsymbol{X}^c$ | Environment invariant features |
| $\boldsymbol{X}^v$ | Environment specific features |
| $Y$ | Response |
| $T$ | Incentive |
| $\Phi(\cdot)$ | The basic backbone of feature encoder |
| $do(\cdot)$ | Intervention operation |
| $h(\cdot)$ | Response prediction head |
| $SF(\cdot, \cdot)$ | Sufficient term |
| $NC(\cdot, \cdot)$ | Necessary term |
| $\Theta$ | Parameters of the sufficient term |
| $\Psi$ | Parameters of the necessary term |
| $\beta_q(\cdot \| \cdot)$ | $\beta$ divergence between environments |
| $\pi_{\boldsymbol{X}}$ | Prior distribution of $\boldsymbol{X}$ |
| $\xi_{e' \backslash e}(\cdot)$ | The worst risk in unknown area |
| $M_e^h(\cdot)$ | Monotonicity measurement |
| $\mathbb{E}$ | Expectation |
| $m(\cdot)$ | Mask vector |
| $\mathrm{KL}(\cdot \mid \cdot)$ | KL divergence |
| $\odot$ | Element-wise multiplication |

We present the detailed description of the metrics we used in the following:

- AUUC (AREA UNDER UPLIFT CURVE): A common metric to evaluate the area under the uplift curve (Rzepakowski & Jaroszewicz, 2010). We use the CausalML package (Chen et al., 2020) to implement the metric.

- QINI (QINI COEFFICIENT) (Mouloud et al., 2020): A common metric to evaluate the area under the qini curve, different from AUUC, it scale the responses in control group.

- KENDALL (KENDALL'S RANK CORRELATION) (Mouloud et al., 2020): A metric to evaluate the average predicted uplift and the predicted uplift in each bin, we report the result of 20 bins.

And for the reading clarity, we provide a notation table in the following.

### A.3. Computational Complexity

In this section, we show the computational complexity in the following.

For the invariant property learning module, the computational complexity of the objective loss and negative log-likelihood is linear, denoted as $O(n)$. The complexity of the KL divergence, however, is $O(n \times d)$, where $d$ represents the feature dimensionality. For the feature selection module, the computational complexity of the linear transformation and batch normalization in forward propagation is $O(n \times h)$, where $n$ is the input dimensionality and $h$ is the dimension of the hidden layer. In the mask generation phase, the Gumbel-Softmax operation is applied iteratively $k$ times, resulting in a complexity of $O(k \times m)$, where $m$ is the number of mask elements. Consequently, the overall complexity of the mask generator can be expressed as $O(n \times h + k \times m)$. For the discrepancy balancing module, for the computational complexity of IPM, if MMD distance is employed, the complexity increases to $O(n^2)$. In summary, the overall time complexity of the algorithm is $O(n^2) + O(n \times h) + O(k \times m)$.

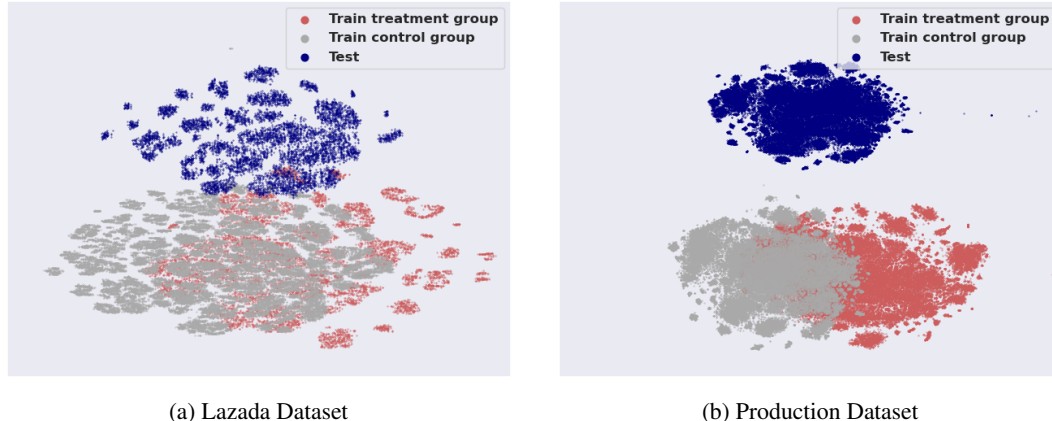

| (a) Lazada Dataset | (b) Production Dataset |

*Figure 5.* Visualization of dataset distribution for in-distribution and out-of-distribution.

### A.4. Analytical Experiments

To better help readers understand our motivation, *i.e.*, the necessity of addressing selection bias and OOD problems in a unified way, we visualize the distribution of the training set (including treatment and control groups) and the test set of the dataset used in the experiment. This is shown in Figure 5, which illustrates the distribution gaps found by our model. Combining the results in Tables 1 and 2, we see that this property makes our model more generalizable than existing baselines under different test settings.

### A.5. Online Experiment

To further show the effectiveness of our IDUM, we conduct an online experiment to evaluate our IDUM against the baseline CFRNet, and we keep it for three days.

The experimental setups are listed in the following:

- **In-Distribution (ID) Test:** We selected two user groups with similar distributions to the training data.

- **Out-of-Distribution (OOD) Test:** We used two user groups with distinct distributions to assess generalization.

Due to the constraints of online inference, we could not obtain the true responses for users unaffected by the deployed algorithm. Instead, we compared the watch time improvement and the cost reduction across experimental groups as the key metrics.

*Table 6.* Online experimental results.

(a) Watch Time Improvement (%)

| Method | ID | OOD |
| --- | --- | --- |
| CFRNet | 0 | 0 |
| IDUM | 0.012% | 0.028% |

(b) Cost Reduction (%)

| Method | ID | OOD |
| --- | --- | --- |
| CFRNet | 0 | 0 |
| IDUM | -1.21% | -1.75% |

We report the percentage of watch time improvement relative to CFRNet's performance (baseline, set at 0). And also, we evaluate the cost of two methods, for that we assign less incentives ($-1.21\%$ and $-1.75\%$) to the user group and get competitive performance ($0.012\%$ and $0.028\%$).

# B. Proofs

## B.1. Proof of Proposition 4.5

To explicitly define the Monotonicity evaluator, we decompose the original objective into three terms: the sufficiency objective $SF_{e'}(h, \Theta)$, the necessity objective $NC_{e'}(h, \Psi)$, and the Monotonicity evaluator objective $M_{e'}^h(\Theta, \Psi)$. The Monotonicity term $M_{e'}^h(\Theta, \Psi)$ can be further decomposed as follows:

$$M_{e'}^h(\Theta, \Psi) = SF_{e'}(h, \Theta)\left(1 - NC_{e'}(h, \Psi)\right) + \left(1 - SF_{e'}(h, \Theta)\right) NC_{e'}(h, \Psi). \tag{16}$$

We can further derive Eq. (16) as follows.

$$
\begin{aligned}
M_{e'}^h(\Theta, \Psi) &= SF_{e'}(h, \Theta)\left(1 - NC_{e'}(h, \Psi)\right) + \left(1 - SF_{e'}(h, \Theta)\right) NC_{e'}(h, \Psi) \\
&= SF_{e'}(h, \Theta) + NC_{e'}(h, \Psi) - 2SF_{e'}(h, \Theta)NC_{e'}(h, \Psi) \\
&= R_{e'}(h, \Theta, \Psi) - 2SF_{e'}(h, \Theta)NC_{e'}(h, \Psi).
\end{aligned} \tag{17}
$$

Then we can rewrite the original objective Eq. (6) by

$$R_{e'}(h, \Theta, \Psi) = M_{e'}^h(\Theta, \Psi) + 2SF_{e'}(h, \Theta)NC_{e'}(h, \Psi).$$

## B.2. Proof of Theorem 4.6

The rationale behind Theorem 4.6 is to establish a connection between the risk observed in the source environment and that in the target environment. To prove the result stated in Theorem 4.6, we make reference to the technical details presented in (Germain et al., 2016). We initially define $u = \mathbb{E}_{(\boldsymbol{x}, t, y) \sim e'} \mathbb{I}[(\boldsymbol{x}, t, y) \notin \sup(e)]$, allowing us to compute the value of $\delta$-Monotonicity measurement on samples originating from the target environment but not encompassed within the source environment:

$$
\begin{aligned}
&\mathbb{E}_{(\boldsymbol{x}, t, y) \sim e'} \mathbb{I}[(\boldsymbol{x}, t, y) \notin \operatorname{supp}(e)] \mathbb{E}_{\boldsymbol{x}^c \sim P_{e'}^\Theta(\boldsymbol{X}^c | \Phi(\boldsymbol{X}) = \Phi(\boldsymbol{x}))} \\
&\mathbb{E}_{\overline{\boldsymbol{x}}^c \sim P_{e'}^\Psi(\boldsymbol{X}^c | \Phi(\boldsymbol{X}) = \Phi(\boldsymbol{x}))} \mathbb{I}\left[h(\boldsymbol{x}^c, t) \neq h(\overline{\boldsymbol{x}}^c, t)\right] \\
=&u \mathbb{E}_{e' \setminus e} \mathbb{E}_{\boldsymbol{x}^c \sim P_{e'}^\Theta(\boldsymbol{X}^c | \Phi(\boldsymbol{X}) = \Phi(\boldsymbol{x}))} \mathbb{E}_{\overline{\boldsymbol{x}}^c \sim P_{e'}^\Psi(\overline{\boldsymbol{X}}^c | \Phi(\boldsymbol{X}) = \Phi(\boldsymbol{x}))} \\
&\mathbb{I}\left[h(\boldsymbol{x}^c, t) \neq h(\overline{\boldsymbol{x}}^c, t)\right] \\
=&u M_{e'}^h(\Theta, \Psi).
\end{aligned}
$$

Likewise, the aggregate risk associated with samples originating from the target environment but not encompassed within the source environment is denoted as $\xi_{e' \setminus e}(\boldsymbol{x}, t, y)$. Next, we proceed to alter the distribution measure, taking $M_{e'}^h(\Theta, \Psi)$ from Eq. (8) as an illustrative example.

$$
\begin{aligned}
&M_{e'}^h(\Theta, \Psi) \\
=&\mathbb{E}_{(\boldsymbol{x}, t, y) \sim e'} \mathbb{E}_{\boldsymbol{x}^c \sim P_{e'}^\Theta(\boldsymbol{X}^c | \Phi(\boldsymbol{X}) = \Phi(\boldsymbol{x}))} \mathbb{E}_{\overline{\boldsymbol{x}}^c \sim P_{e'}^\Psi(\overline{\boldsymbol{X}}^c | \Phi(\boldsymbol{X}) = \Phi(\boldsymbol{x}))} \\
&\mathbb{I}\left[h(\boldsymbol{x}^c, t) \neq h(\overline{\boldsymbol{x}}^c, t)\right] \\
\leq&\beta_q(e' \| e) \left[\mathbb{E}_{\boldsymbol{x}^c \sim P_e^\Theta(\boldsymbol{X}^c | \Phi(\boldsymbol{X}) = \Phi(\boldsymbol{x}))} \mathbb{E}_{\overline{\boldsymbol{x}}^c \sim P_e^\Psi(\overline{\boldsymbol{X}}^c | \Phi(\boldsymbol{X}) = \Phi(\boldsymbol{x}))}\right. \\
&\left. \mathbb{I}\left[h(\boldsymbol{x}^c, t) \neq h(\overline{\boldsymbol{x}}^c, t)\right]^{\frac{q}{q-1}}\right]^{1 - \frac{1}{q}} + u M_{e' \setminus e}^h(\Theta, \Psi) \\
=&\beta_q(e' \| e) \left[\mathbb{E}_{\boldsymbol{x}^c \sim P_e^\Theta(\boldsymbol{X}^c | \Phi(\boldsymbol{X}) = \Phi(\boldsymbol{x}))} \mathbb{E}_{\overline{\boldsymbol{x}}^c \sim P_e^\Psi(\overline{\boldsymbol{X}}^c | \Phi(\boldsymbol{X}) = \Phi(\boldsymbol{x}))}\right. \\
&\left. \mathbb{I}\left[h(\boldsymbol{x}^c, t) \neq h(\overline{\boldsymbol{x}}^c, t)\right]\right]^{1 - \frac{1}{k}} + u M_{e' \setminus e}^h(\Theta, \Psi).
\end{aligned}
$$

The third line is a consequence of Hölder's inequality. In the final line, we omit the exponential term $\frac{q}{q-1}$ in $\mathbb{I}[h(\boldsymbol{x}^c, t) \neq h(\overline{\boldsymbol{x}}^c, t)]^{\frac{q}{q-1}}$ as the function yields results from the set $\{0, 1\}$. Similarly, for the term $SF_e(h, \Theta)NC_{e'}(h, \Psi)$, we derive the ultimate bound for the overall $R_{e'}(h, \Theta, \Psi)$ as shown in Eq. (8).

## B.3. Proof of Theorem 4.7

In this theorem, we investigate the bounding of distribution risk by empirical risk. The proof employs well-known inequalities such as Jensen's inequality, Markov's inequality, and Hoeffding's inequality. We initially focus on the term $SF_e(h, \Theta)$. The outline of the proof involves utilizing variational inference techniques to alter the distribution measure, followed by the application of Markov's inequality to compute the risk bound.

Let $\Delta\left(SF_e(h, \Theta)\right) = \widetilde{SF}_e(h, \Theta) - SF_e(h, \Theta)$. By employing the variational inference technique and leveraging Jensen's inequality, we derive the following inequality:

$$
\begin{aligned}
&4(n-1)^2 \Delta\left(SF_e(h, \Theta)\right)^2 \\
&= 4(n-1)^2 \left(\widetilde{SF}_e(h, \Theta) - SF_e(h, \Theta)\right)^2 \\
&= \left(2(n-1)\mathbb{E}_{e^n}\mathrm{KL}\left(\tilde{P}_s^\phi(\boldsymbol{X}^c \mid \Phi(\boldsymbol{X}) = \Phi(\boldsymbol{x})) \| \pi_{\boldsymbol{X}^c}\right)\right. \\
&\quad - 2(n-1)\mathbb{E}_e \mathbb{E}_{P_e^\Theta(\boldsymbol{X}^c \mid \Phi(\boldsymbol{X})=\Phi(\boldsymbol{x}))} \ln \frac{\pi_{\boldsymbol{X}^c}}{P_e^\Theta(\boldsymbol{X}^c \mid \Phi(\boldsymbol{X}) = \Phi(\boldsymbol{x}))} \\
&\quad + \mathbb{E}_{\boldsymbol{x}^c \sim \pi_{\boldsymbol{X}^c}} \ln \exp\left(2(n-1)\mathbb{E}_{e^n}\mathbb{I}\left[h\left(\boldsymbol{x}^c, t\right) \neq y\right]\right) \\
&\quad - \mathbb{E}_{\boldsymbol{x}^c \sim \pi_{\boldsymbol{X}^c}} \ln \exp\left(2(n-1)\mathbb{E}_e \mathbb{I}\left[h\left(\boldsymbol{x}^c, t\right) \neq y\right]\right)^2, \\
&\leq \left(2(n-1)\mathbb{E}_{e^n}\mathrm{KL}\left(\tilde{P}_e^\Theta(\boldsymbol{X}^c \mid \Phi(\boldsymbol{X}) = \Phi(\boldsymbol{x})) \| \pi_{\boldsymbol{X}^c}\right)\right. \\
&\quad - 2(n-1)\mathbb{E}_e \mathbb{E}_{P_e^\Theta(\boldsymbol{X}^c \mid \Phi(\boldsymbol{X})=\Phi(\boldsymbol{x}))} \ln \frac{\pi_{\boldsymbol{X}^c}}{P_e^\Theta(\boldsymbol{X}^c \mid \Phi(\boldsymbol{X}) = \Phi(\boldsymbol{x}))} \\
&\quad + \ln \mathbb{E}_{\boldsymbol{x}^c \sim \pi_{\boldsymbol{X}^c}} \exp\left(2(n-1)\Delta\left(SF_e'\right)\right)^2, \\
&\leq \left(2(n-1)\mathbb{E}_{e^n}\mathrm{KL}(q(\boldsymbol{x}^c \mid \Phi(\boldsymbol{x})) \| p(\boldsymbol{X}^c))\right. \\
&\quad - 2(n-1)\mathbb{E}_e \mathrm{KL}\left(P_e^\Theta(\boldsymbol{X}^c \mid \Phi(\boldsymbol{X}) = \Phi(\boldsymbol{x})) \| \pi_{\boldsymbol{X}^c}\right) \\
&\quad + \ln \mathbb{E}_{\boldsymbol{x}^c \sim p(\boldsymbol{X}^c)} \exp\left(2(n-1)\Delta\left(SF_e'\right)\right)^2,
\end{aligned}
\tag{18}
$$

where $\Delta\left(SF_e'\right) = \left|\mathbb{E}_{e^n}\mathbb{I}\left[h\left(\boldsymbol{x}^c, t\right) \neq y\right] - \mathbb{E}_e \mathbb{I}\left[h\left(\boldsymbol{x}^c, t\right) \neq y\right]\right|$. Recall that Hoeffding's inequality, we get the following inequality.

$$
P_{(e)^{2n-1}}\left[\Delta\left(SF_e'\right) \geq \eta\right] \leq \exp(-2n)\eta^2
\tag{19}
$$

Then, denoting the density function of $\Delta\left(SF_e'\right)$ as $f\left(\Delta\left(SF_e'\right)\right)$ )

$$
\begin{aligned}
&P_{(\mathcal{SF})^{2n-1}}\left[\Delta\left(SF_e'\right) \geq \eta\right] = e^{-2n\eta^2} \\
&\Rightarrow \int_\eta^\infty f\left(\Delta\left(SF_e'\right) d\Delta\left(SF_e'\right)\right) = e^{-2m\eta^2} \\
&\Rightarrow f(\eta) = 4m\eta e^{-2m\eta^2}.
\end{aligned}
$$

Then, we get

$$
\begin{aligned}
&\mathbb{E}\left[\exp\left(2(n-1)\Delta\left(SF_e'\right)\right)\right] \\
&= \int_0^1 f\left(\Delta\left(SF_e'\right)\right) \exp\left(2(n-1)\Delta\left(SF_e'\right)\right) d\Delta\left(SF_e'\right) \\
&\leq \int_0^1 4n\Delta\left(SF_e'\right) \exp\left(-2n\Delta\left(SF_e'\right)\right) \exp\left(2(n-1)\Delta\left(SF_e'\right)\right) d\Delta\left(SF_e'\right) \\
&= n \int_0^1 2\Delta\left(SF_e'\right) \exp\left(-2\Delta\left(SF_e'\right)\right) d2\Delta\left(SF_e'\right) \\
&= -\left. ne^{-2\Delta\left(SF_e'\right)}\left(2\Delta\left(SF_e'\right) + 1\right)\right|_0^1 \\
&< -\left. ne^{-2\Delta\left(SF_e'\right)}\left(2\Delta\left(SF_e'\right) + 1\right)\right|_0^\infty \\
&= n \lim_{\Delta(SF_e') \to \infty} -e^{-2\Delta\left(SF_e'\right)}\left(2\Delta\left(SF_e'\right) + 1\right) + n = n.
\end{aligned}
\tag{20}
$$

Combining Eq. (20) with Eq. (18), by Markov's inequality, we further get:

$$P_{(e)^{2n-1}}\left[\Delta\left(SF'_e\right) \geq \eta\right] \leq \frac{n}{e^{\eta}}$$

Suppose that $\eta = \ln(n/\epsilon)$, and with the probability of at least $1 - \epsilon$, we have that for all $\pi_{\boldsymbol{X}^c}$,

$$
\begin{aligned}
&Eq.~(18)\\
&\leq \Big(2(n-1)\mathbb{E}_{e^n}\mathrm{KL}\left(\tilde{P}_e^{\Theta}(\boldsymbol{X}^c \mid \Phi(\boldsymbol{X}) = \Phi(\boldsymbol{x}))\|\pi_{\boldsymbol{X}^c}\right)\\
&\quad - 2(n-1)\mathbb{E}_e\mathrm{KL}\left(P_e^{\Theta}(\boldsymbol{X}^c \mid \Phi(\boldsymbol{X}) = \Phi(\boldsymbol{x}))\|\pi_{\boldsymbol{X}^c}\right) + \ln(n/\epsilon)\Big)^2
\end{aligned}
$$

Then we have

$$
\begin{aligned}
&\left|\widetilde{SF}_e(h, \Theta) - SF_e(h, \Theta)\right|\\
&\leq \left|\mathbb{E}_{e^n}\mathrm{KL}\left(\tilde{P}_e^{\Theta}(\boldsymbol{X}^c \mid \Phi(\boldsymbol{X}) = \Phi(\boldsymbol{x}))\|\pi_{\boldsymbol{X}^c}\right)\right.\\
&\quad \left. -\mathbb{E}_e\mathrm{KL}\left(P_e^{\Theta}(\boldsymbol{X}^c \mid \Phi(\boldsymbol{X}) = \Phi(\boldsymbol{x}))\|\pi_{\boldsymbol{X}^c}\right) + \frac{1}{2(n-1)}\ln(n/\epsilon)\right|\\
&\leq \left|\mathbb{E}_{e^n}\mathrm{KL}\left(\tilde{P}_e^{\Theta}(\boldsymbol{X}^c \mid \Phi(\boldsymbol{X}) = \Phi(\boldsymbol{x}))\|\pi_{\boldsymbol{X}^c}\right)\right.\\
&\quad \left. +\mathbb{E}_e\mathrm{KL}\left(P_e^{\Theta}(\boldsymbol{X}^c \mid \Phi(\boldsymbol{X}) = \Phi(\boldsymbol{x}))\|\pi_{\boldsymbol{X}^c}\right) + \frac{1}{2(n-1)}\ln(n/\epsilon)\right|
\end{aligned}
$$

According to the assumption, we have

$$
\begin{aligned}
&\left|\widetilde{SF}_e(h, \Theta) - SF_e(h, \Theta)\right|\\
&\leq \mathbb{E}_{e^n}\mathrm{KL}\left(\tilde{P}_e^{\Theta}(\boldsymbol{X}^c \mid \Phi(\boldsymbol{X}) = \Phi(\boldsymbol{x}))\|\pi_{\boldsymbol{X}^c}\right) + \frac{1}{2(n-1)}\ln(n/\epsilon) + C.
\end{aligned}
$$

We get the results demonstrated in Theorem 4.7 (1).

For the term $M_e^h(\Theta, \Psi)$, we define $\Delta(M_e) = M_e^h(\Theta, \Psi) - \widetilde{M}_e^h(\Theta, \Psi)$, where $\widetilde{M}_e^h(\Theta, \Psi) := \mathbb{E}_{e^n}\mathbb{E}_{\boldsymbol{x}^c \sim \tilde{P}_e^{\Theta}(\boldsymbol{X}^c|\Phi(\boldsymbol{X})=\Phi(\boldsymbol{x}))}\mathbb{E}_{\overline{\boldsymbol{x}}^c \sim \tilde{P}_e^{\Psi}(\overline{\boldsymbol{X}}^c|\Phi(\boldsymbol{X})=\Phi(\boldsymbol{x}))} \mathbb{I}\left[h\left(\boldsymbol{x}^c, t\right) \neq h\left(\overline{\boldsymbol{x}}^c, t\right)\right]$. Different from Theorem 4.7 (1), the monotonicity measurement has an extra expectation on $\overline{\boldsymbol{X}}^c$. We apply Jensen's inequality again and then use the variational inference trick to get the derivation results.

$$4(n-1)^2\Delta(M_e)^2 = 4(n-1)^2\left(M_e^h(\Theta, \Psi) - \widetilde{M}_e^h(\Theta, \Psi)\right)^2 \tag{21}$$

Then, we consider the term $M_e^h(\Theta, \Psi)$ and $\widetilde{M}_e^h(\Theta, \Psi)$ separately.

$$
\begin{aligned}
&M_e^h(\Theta, \Psi)\\
&=\mathbb{E}_{e^n}\mathbb{E}_{\boldsymbol{x}^c \sim P_e^{\Theta}(\boldsymbol{X}^c|\Phi(\boldsymbol{X})=\Phi(\boldsymbol{x}))}\mathbb{E}_{\overline{\boldsymbol{x}}^c \sim P_e^{\Psi}(\overline{\boldsymbol{X}}^c|\Phi(\boldsymbol{X})=\Phi(\boldsymbol{x}))}\mathbb{I}\left[h\left(\boldsymbol{x}^c, t\right) \neq h\left(\overline{\boldsymbol{x}}^c, t\right)\right]\\
&=\mathbb{E}_e\left[\mathbb{E}_{\boldsymbol{X}^c}\mathbb{E}_{\overline{\boldsymbol{X}}^c}\ln\frac{P_e^{\Theta}(\boldsymbol{X}^c \mid \Phi(\boldsymbol{X}) = \Phi(\boldsymbol{x}))}{\pi_{\boldsymbol{X}^c}} + \mathbb{E}_{\boldsymbol{X}^c}\mathbb{E}_{\overline{\boldsymbol{X}}^c}\right.\\
&\quad \ln\frac{P_e^{\Psi}(\overline{\boldsymbol{X}}^c \mid \Phi(\boldsymbol{X}) = \Phi(\boldsymbol{x}))}{\pi_{\overline{\boldsymbol{X}}^c}}\\
&\quad +\mathbb{E}_{\boldsymbol{X}^c}\mathbb{E}_{\overline{\boldsymbol{X}}^c}\ln\frac{\pi_{\boldsymbol{X}^c}}{P_e^{\Theta}(\boldsymbol{X}^c \mid \Phi(\boldsymbol{X}) = \Phi(\boldsymbol{x}))}\frac{\pi_{\overline{\boldsymbol{X}}^c}}{P_e^{\Psi}(\overline{\boldsymbol{X}}^c \mid \Phi(\boldsymbol{X}) = \Phi(\boldsymbol{x}))}\\
&\quad \exp\left(\mathbb{I}\left[h\left(\boldsymbol{x}^c, t\right) \neq h\left(\overline{\boldsymbol{x}}^c, t\right)\right]\right)]\\
&\leq \mathbb{E}_{(\boldsymbol{x}, t, y) \sim e}\left[\mathrm{KL}\left(P_e^{\Theta}(\boldsymbol{X}^c \mid \Phi(\boldsymbol{X}) = \Phi(\boldsymbol{x}))\|\pi_{\boldsymbol{X}^c}\right)\right.\\
&\quad \left. +\mathrm{KL}\left(P_e^{\Psi}(\overline{\boldsymbol{X}}^c \mid \Phi(\boldsymbol{X}) = \Phi(\boldsymbol{x}))\|\pi_{\overline{\boldsymbol{X}}^c}\right)\right]\\
&\quad +\ln\mathbb{E}_{\boldsymbol{x}^c \sim \pi_{\boldsymbol{X}^c}}\mathbb{E}_{\overline{\boldsymbol{x}}^c \sim \pi_{\overline{\boldsymbol{X}}^c}}\exp\left(\mathbb{E}_e\mathbb{I}\left[h\left(\boldsymbol{x}^c, t\right) \neq h\left(\overline{\boldsymbol{x}}^c, t\right)\right]\right)
\end{aligned}
\tag{22}
$$

Similarly, for the empirical risk $\widetilde{M}_e^h(\Theta, \Psi)$:

$$
\begin{aligned}
&\widetilde{M}_e^h(\Theta, \Psi)\\
=&\mathbb{E}_{(\boldsymbol{x},t,y)\sim e^n}\mathbb{E}_{\boldsymbol{x}^c\sim\tilde{P}_e^\Theta(\boldsymbol{X}^c|\Phi(\boldsymbol{X})=\Phi(\boldsymbol{x}))}\mathbb{E}_{\overline{\boldsymbol{x}}^c\sim\tilde{P}_e^\Psi(\overline{\boldsymbol{X}}^c|\Phi(\boldsymbol{X})=\Phi(\boldsymbol{x}))}\\
&\quad\mathbb{I}\left[h\left(\boldsymbol{x}^c, t\right)\neq h\left(\overline{\boldsymbol{x}}^c, t\right)\right]\\
=&\mathbb{E}_e\left[\mathbb{E}_{\boldsymbol{X}^c}\mathbb{E}_{\overline{\boldsymbol{X}}^c}\ln\frac{\tilde{P}_e^\Theta(\boldsymbol{X}^c\mid\Phi(\boldsymbol{X})=\Phi(\boldsymbol{x}))}{\pi_{\boldsymbol{X}^c}}+\mathbb{E}_{\boldsymbol{X}^c}\mathbb{E}_{\overline{\boldsymbol{X}}^c}\right.\\
&\quad\ln\frac{\tilde{P}_e^\Psi(\overline{\boldsymbol{X}}^c\mid\Phi(\boldsymbol{X})=\Phi(\boldsymbol{x}))}{\pi_{\overline{\boldsymbol{X}}^c}}\\
&\quad+\mathbb{E}_{\boldsymbol{X}^c}\mathbb{E}_{\overline{\boldsymbol{X}}^c}\ln\frac{\pi_{\boldsymbol{X}^c}}{\tilde{P}_e^\Theta(\boldsymbol{X}^c\mid\Phi(\boldsymbol{X})=\Phi(\boldsymbol{x}))}\frac{\pi_{\overline{\boldsymbol{X}}^c}}{\tilde{P}_e^\Psi(\overline{\boldsymbol{X}}^c\mid\Phi(\boldsymbol{X})=\Phi(\boldsymbol{x}))}\\
&\quad\left.\exp\left(\mathbb{I}\left[h\left(\boldsymbol{x}^c, t\right)\neq h\left(\overline{\boldsymbol{x}}^c, t\right)\right]\right)\right]\\
\leq&\mathbb{E}_{(\boldsymbol{x},t,y)\sim e}\left[\text{KL}\left(\tilde{P}_e^\Theta(\boldsymbol{X}^c\mid\Phi(\boldsymbol{X})=\Phi(\boldsymbol{x}))\|\pi_{\boldsymbol{X}^c}\right)\right.\\
&\quad\left.+\text{KL}\left(\tilde{P}_e^\Psi(\overline{\boldsymbol{X}}^c\mid\Phi(\boldsymbol{X})=\Phi(\boldsymbol{x}))\|\pi_{\overline{\boldsymbol{X}}^c}\right)\right]\\
&\quad+\ln\mathbb{E}_{\boldsymbol{x}^c\sim\pi_{\boldsymbol{X}^c}}\mathbb{E}_{\overline{\boldsymbol{x}}^c\sim\pi_{\overline{\boldsymbol{X}}^c}}\exp\left(\mathbb{E}_e\mathbb{I}\left[h\left(\boldsymbol{x}^c, t\right)\neq h\left(\overline{\boldsymbol{x}}^c, t\right)\right]\right).
\end{aligned}
\tag{23}
$$

Combining Eq. (23) with Eq. (22) and plugin to Eq. (21), we have

$$
\begin{aligned}
&4(n-1)^2\Delta\left(M_e\right)^2\\
=&4(n-1)^2\left(M_e^h(\Theta, \Psi)-\widetilde{M}_e^h(\Theta, \Psi)\right)^2\\
\leq&\left(2(n-1)\left(\mathbb{E}_{e^n}\text{KL}\left(\tilde{P}_e^\Theta(\boldsymbol{X}^c\mid\Phi(\boldsymbol{X})=\Phi(\boldsymbol{x}))\|\pi_{\boldsymbol{X}^c}\right)\right.\right.\\
&\quad+\mathbb{E}_{e^n}\text{KL}\left(\tilde{P}_e^\Psi(\overline{\boldsymbol{X}}^c\mid\Phi(\boldsymbol{X})=\Phi(\boldsymbol{x}))\|\pi_{\overline{\boldsymbol{X}}^c}\right)\\
&\quad\left.+\mathbb{E}_e\text{KL}\left(\tilde{P}_e^\Psi(\overline{\boldsymbol{X}}^c\mid\Phi(\boldsymbol{X})=\Phi(\boldsymbol{x}))\|\pi_{\overline{\boldsymbol{X}}^c}\right)\right)\\
&\quad\left.+\ln\mathbb{E}_{\boldsymbol{x}^c\sim\pi_{\boldsymbol{X}^c}}\mathbb{E}_{\boldsymbol{x}^c\sim\pi_{\overline{\boldsymbol{X}}^c}}\exp\left(2(n-1)\Delta\left(M_e'\right)\right)\right)^2,
\end{aligned}
$$

where $\Delta\left(M_e'\right)=\left|\mathbb{E}_{e^n}\mathbb{I}\left[h\left(\boldsymbol{x}^c, t\right)\neq h\left(\overline{\boldsymbol{x}}^c, t\right)\right]-\mathbb{E}_e\mathbb{I}\left[h\left(\boldsymbol{x}^c, t\right)\neq h\left(\overline{\boldsymbol{x}}^c, t\right)\right]\right|$. The rest of the proof is similar to Theorem 4.7 (1), thus we get the theoretical results of Theorem 4.7 (2).

