# OpenReview forum: "Invariant Deep Uplift Modeling for Incentive Assignment in Online Marketing via Probability of Necessity and Sufficiency"
_ICML.cc/2025/Conference — ICML 2025 spotlightposter_

### Official Review · Reviewer_UCNa · 2025-03-10

**Overall Recommendation:** 4

**Summary:**

This work investigates uplift modeling for incentive allocation in digital marketplaces, addressing two critical challenges: OOD generalization and selection bias mitigation through spurious correlation elimination. The proposed IDUM framework introduces: 1) A causal invariance learning mechanism identifying domain-agnostic features with necessity and sufficiency properties. 2) A differentiable Gumbel-softmax feature selector for computational efficiency. 3) A distributional discrepancy regularizer aligning treatment-control group representations. Empirical validation on production datasets demonstrates IDUM's superior performance in response prediction compared to existing baselines.

**Claims And Evidence:**

Yes.

**Essential References Not Discussed:**

No.

**Experimental Designs Or Analyses:**

Yes.

**Methods And Evaluation Criteria:**

Yes.

**Other Comments Or Suggestions:**

No.

**Other Strengths And Weaknesses:**

**Strengths**

Methodological Innovation: IDUM presents a novel paradigm for uplift modeling under distribution shifts. Its emphasis on causal invariance and feature sufficiency/necessity decomposition addresses critical gaps in real-world deployment scenarios.
​Theoretical Rigor: The authors establish formal generalization guarantees through β-divergence analysis and PNS risk bounding. The integration of distributional alignment metrics (IPM, MMD) provides principled control of treatment group biases.
Good narration: The authors skillfully present the main ideas of the article, helping to understand the key counter-attributions.

**Weaknesses**

1. The mathematical formulations underpinning the proposed methodology present certain complexities in comprehension, which could be mitigated through supplementary elucidation to strengthen theoretical transparency.
2. The work lacks an explicit discussion of the model's computational overhead during training phases and omits details regarding algorithmic complexity analysis. These omissions hinder the practical evaluation of real-world deployment feasibility. Furthermore, the absence of an exhaustive specification of architectural hyperparameters and optimization configurations compromises the reproducibility of experimental results across diverse datasets.

**Questions For Authors:**

Please see "Weaknesses"

**Relation To Broader Scientific Literature:**

This paper addresses the challenge of testing data that is out-of-distribution.

**Theoretical Claims:**

Yes.

---

> ### Author Rebuttal · Authors · 2025-03-31
>
> Dear Reviewer UCNa,
>
> Thank you for taking the time to review our work, we sincerely appreciate your insightful comments, which have helped improve our paper. Below, we provide point-by-point responses to each concern raised.
>
> 1. **Mathematical Formulations**: We appreciate the valuable feedback and have incorporated a comprehensive notation table in the appendix to improve clarity. In our final revision, we will further enhance these explanations to ensure better understanding of our methodology.
>
> 2. **Computational Complexity Analysis**: We analyze the computational complexity of each module in our framework:
>
> **Invariant Property Learning Module**
>    - Objective loss and negative log-likelihood: $O(n)$ (linear complexity)
>    - KL divergence: $O(n \times d)$, where $d$ is the feature dimensionality
>
> **Feature Selection Module**
>    - Forward propagation (linear transformation + batch normalization): $O(n \times h)$, where $h$ is hidden layer dimension
>    - Mask generation (Gumbel-Softmax): $O(k \times m)$ for $k$ iterations on $m$ mask elements
>    - Total complexity: $O(n \times h + k \times m)$
>
> **Discrepancy Balancing Module**
>    - IPM with MMD distance: $O(n^2)$
>
> The overall time complexity of IDUM is $O(n^2) + O(n \times h) + O(k \times m)$.
>
> **Empirical Runtime Comparison**
>
> We compare the computational efficiency of different methods on two datasets:
>
> | Method         | Production Dataset | Lazada Dataset |
> |----------------|-------------------:|---------------:|
> | S-Learner      | 37 min            | 27 min         |
> | T-Learner      | 40 min            | 28 min         |
> | TARNet         | 58 min            | 49 min         |
> | CFRNet-mmd     | 61 min            | 53 min         |
> | CFRNet-wass    | 122 min           | 117 min        |
> | DragonNet      | 73 min            | 64 min         |
> | EUEN           | 65 min            | 35 min         |
> | UniTE          | 110 min           | 86 min         |
> | TEED           | 93 min            | 89 min         |
> | IDUM (ours)    | 85 min            | 79 min         |
>
> If you have any further questions to discuss, we are willing to reply as soon as possible.

---

### Official Review · Reviewer_4KLk · 2025-03-12

**Overall Recommendation:** 4

**Summary:**

This study addresses uplift modeling for incentive allocation in online markets, aiming to resolve out-of-distribution generalization challenges and selection bias by eliminating spurious correlations. The authors propose the Invariant Deep Uplift Modeling (IDUM) framework, which innovatively integrates a cross-domain invariant learning approach to identify causal features with necessary and sufficient properties. The model employs a Gumbel-Softmax-based feature selection module to enhance computational efficiency and introduces a balanced discrepancy constraint mechanism to mitigate distributional biases between treatment and control groups. Empirical evaluations on real-world commercial datasets (e.g., Lazada) demonstrate that the proposed method significantly outperforms existing benchmarks in response prediction tasks, showcasing superior causal inference capabilities. Overall, the paper exhibits clear motivations, a rigorous theoretical framework, and well-designed experiments with practical implications, all presented with logical coherence and academic rigor.

**Claims And Evidence:**

See summary.

**Essential References Not Discussed:**

See summary.

**Experimental Designs Or Analyses:**

See summary.

**Methods And Evaluation Criteria:**

See summary.

**Other Comments Or Suggestions:**

NA

**Other Strengths And Weaknesses:**

Pros:
1. The IDUM framework introduces an improved approach to overcome limitations in handling out-of-distribution (OOD) challenges. Its design consists of three straightforward components: invariant property learning, feature selection, and balancing discrepancy. The method has been carefully developed and supported by mathematical analysis.
2. Extensive experiments are performed to validate the model's effectiveness. Experiments covered two widely-used benchmark datasets and one real-world industrial dataset, with performance comparisons made against standard reference methods in the field.

Cons:
1. The study currently does not publicly share its implementation code, which limits reproducibility and poses a challenge for independent verification or further development by the research community.
2. Some technical sections involving mathematical formulations may require simplifications or clearer explanations to improve accessibility for readers with varying levels of expertise.

**Questions For Authors:**

NA

**Relation To Broader Scientific Literature:**

See summary.

**Theoretical Claims:**

See summary.

---

> ### Author Rebuttal · Authors · 2025-03-31
>
> **Dear Reviewer 4KLk**,
>
> Thank you for taking the time to review our work, we sincerely appreciate your insightful comments, which have helped improve our paper. Below, we provide point-by-point responses to each concern raised.
>
> **Cons**
>
> 1. **Open-source code**: We have included our code in the Supplementary Material for review. Upon acceptance of the paper, we will open-source all code and datasets to ensure reproducibility, and will provide the GitHub link in the final version.
>
> 2. **Mathematical formulation**: Additionally, we have added a notation table in the appendix for clarity, and will further expand the explanations in our final revision to enhance understanding.
>
> If you have any further questions to discuss, we are willing to reply as soon as possible.

---

### Official Review · Reviewer_UDxr · 2025-03-13

**Overall Recommendation:** 4

**Summary:**

The paper proposes Invariant Deep Uplift Modeling (IDUM) for incentive assignment in online marketing (e.g., coupons or discounts). The model identifies features that are both necessary and sufficient under distribution shifts (e.g., changes in user demographics, time, or geography). It builds on the Probability of Necessity and Sufficiency (PNS) framework from Pearl’s causality theory. A Gumbel-Softmax-based selection layer masks out irrelevant or unstable features, enabling the model to focus on “invariant” causal relationships and reduce computational overhead. It uses an integral probability metric (IPM) to control distributional difference to mitigate selection bias, aligning distributions of treated and control users in a latent representation. The experiment shows that IDUM provides robust uplift prediction under out-of-distribution (OOD) shifts and outperforms multiple baselines.

**Claims And Evidence:**

All main claims (better uplift prediction, out-of-distribution robustness, correctness of the bounding arguments) appear to be well-supported by both empirical results and theoretical analysis.

**Essential References Not Discussed:**

NA

**Experimental Designs Or Analyses:**

- The experiments compare IDUM to widely used uplift baselines (S-learner, T-learner, TARNet, CFRNet, DragonNet, etc.) using the same training setup, hyperparameter tuning strategies, and standard metrics.
- Authors test both in-distribution and out-of-distribution performance, which directly addresses the paper’s main claim of robustness to real-world shifts.
- The ablation studies (removing the balancing discrepancy term, the invariant property learning term, or the feature selection) confirm that each piece meaningfully contributes to performance.

**Methods And Evaluation Criteria:**

The proposed method (IDUM) makes sense for the problem: it specifically addresses two big challenges in real-world uplift modeling:
- Selection bias between treated/control groups in observational data.
- Distribution shift over time or across user populations.

The evaluation criteria (AUUC, QINI, Kendall) are standard in uplift modeling literature. The chosen real-world datasets cover ID and OOD scenarios, which is a strong design choice to demonstrate generalization capacity although more diverse evaluation dataset could help prove the model can work well under different conditions.

Overall, the methods align well with the practical online marketing setting, and the metrics/datasets are appropriate and standard for uplift tasks.

**Other Comments Or Suggestions:**

Please see the weakness part.

**Other Strengths And Weaknesses:**

Strengths

- The paper builds its methodology on Pearl’s causality framework, especially around Probability of Necessity and Sufficiency. This is a clear and principled way to identify robust features for uplift.
- The authors convincingly motivate the need for out-of-distribution generalization in real-world incentive assignment and illustrate why traditional uplift approaches can fail under changing user distributions.
-  The paper provides not only an invariant learning approach but also attempts to back it up with a set of domain adaptation–style bounds, which is valuable in a field that often has few formal generalization guarantees.

Weaknesses:

- While the paper does well to include two real-world datasets (Lazada and a short-video platform), these are still both from online platforms. Including more varied or publicly accessible datasets (e.g., from different verticals or domains) could strengthen the claims about broad applicability.
- Their Gumbel-Softmax–based feature selection is a major part of the proposed approach, yet there is only a limited discussion of which features get included or excluded and why. Additional analysis or interpretability results—especially highlighting which features consistently remain selected—could offer insights into the invariant property learning.

**Questions For Authors:**

NA

**Relation To Broader Scientific Literature:**

The approach is well-situated in the intersection of causal inference, domain adaptation, and uplift modeling.
- The paper builds on prior works like S-learner, T-learner, TARNet, CFRNet, etc., addressing the standard challenge of biased observational data and distribution mismatch in real systems.
-  Relates to Pearl’s framework for identifying causal effects under exogeneity and monotonicity. The paper is among the few that directly incorporate “Probability of Necessity and Sufficiency” for robust OOD generalization.
- Connects to literature on distribution shift (e.g., IRM, domain adversarial training), but specifically tailors these ideas for uplift models in marketing contexts.

**Theoretical Claims:**

The theorical claims mostly from Pearl's existing work and are correct. The claims specific for this paper uses bounding techniques consistent with prior works (e.g union bounds, Jensen's inequality), and the proofs given are coherent, consistent with prior literature, and appear correct under the listed assumptions.

---

> ### Author Rebuttal · Authors · 2025-03-31
>
> **Dear Reviewer UDxr**,
>
> Thank you for taking the time to review our work, we sincerely appreciate your insightful comments, which have helped improve our paper. Below, we provide point-by-point responses to each concern raised.
>
> **Weaknesses:**
>
>
>
> 1. **Additional experiments**:
>
> Due to the inherent constraints of our model's design, we are unable to generalize our approach to domains with significantly different characteristics within the rebuttal period. However, in response to your feedback, we have conducted additional experiments on the Criteo dataset (ad click-through rate prediction) to validate our method's ability.
>
> Training Data: 80% (in-distribution)
>
> ID Test Data: 10% (in-distribution)
>
> OOD Test Data: 10% (unbiased sampling)
>
> This partition allows us to evaluate both in-distribution performance and out-of-distribution generalization.
>
>
> | Method          | ID AUUC           | ID QINI           | ID KENDALL       | OOD AUUC          | OOD QINI          | OOD KENDALL     |
> |-----------------|-------------------|-------------------|------------------|-------------------|-------------------|-----------------|
> | S-Learner       | 0.0973 ± 0.0027   | 0.0356 ± 0.0121   | 0.0091 ± 0.0077  | 0.0939 ± 0.0021   | 0.0361 ± 0.0097   | 0.0093 ± 0.0064 |
> | T-Learner       | 0.1025 ± 0.0022   | 0.0422 ± 0.0109   | 0.0117 ± 0.0063  | 0.0919 ± 0.0022   | 0.0411 ± 0.0115   | 0.0087 ± 0.0068 |
> | TARNet          | 0.1031 ± 0.0024   | 0.0411 ± 0.0145   | 0.0109 ± 0.0037  | 0.0915 ± 0.0023   | 0.0398 ± 0.0127   | 0.0093 ± 0.0072 |
> | CFRNet-mmd      | **0.1042 ± 0.0032** | 0.0429 ± 0.0099   | 0.0112 ± 0.0047  | 0.0924 ± 0.0019   | 0.0339 ± 0.0133   | 0.0109 ± 0.0064 |
> | CFRNet-wass     | 0.0922 ± 0.0022   | 0.0391 ± 0.0110   | 0.0122 ± 0.0051  | 0.0908 ± 0.0024   | 0.0401 ± 0.0117   | 0.0101 ± 0.0039 |
> | DragonNet       | 0.1037 ± 0.0020| 0.0429 ± 0.0127   | 0.0112 ± 0.0044  | 0.1009 ± 0.0021   | 0.0372 ± 0.0130   | 0.0107 ± 0.0048 |
> | EUEN            | 0.1029 ± 0.0031   | 0.0421 ± 0.0118   | 0.0132 ± 0.0050  | 0.1007 ± 0.0029   | 0.0396 ± 0.0134   | 0.0092 ± 0.0061 |
> | UniTE           | 0.1033 ± 0.0036   | 0.0467 ± 0.0137   | 0.0157 ± 0.0049  | 0.0901 ± 0.0027   | 0.0401 ± 0.0125   | 0.0073 ± 0.0057 |
> | TEED            | 0.0923 ± 0.0021   | 0.0436 ± 0.0132 | 0.0127 ± 0.0061  | 0.0902 ± 0.0029   | 0.0335 ± 0.0117   | 0.0092 ± 0.0043 |
> | **IDUM**        | 0.1027 ± 0.0023   | __0.0482 ± 0.0093__   | **0.0163 ± 0.0133**  | **0.1019 ± 0.0023** | **0.0442 ± 0.0105** | **0.0112 ± 0.0054** |
>
>
> 2. **Gumble softmax features**:
>
> Both the Lazada dataset (with over 80 features) and the Production dataset (with over 100 features) exhibit high dimensionality. To address the computational challenges posed by such large feature sets, we employ Gumbel-Softmax-based feature selection, which effectively reduces computational costs while preserving predictive performance.
>
> While the Lazada dataset lacks feature interpretability, our analysis of the Production dataset reveals that certain features—such as Phone_type, Region, Silcon, and Age, etc., are consistently retained across selections. This suggests their importance in modeling uplift across diverse scenarios.
>
> If you have any further questions to discuss, we are willing to reply as soon as possible.

---

### Official Review · Reviewer_xxdc · 2025-03-13

**Overall Recommendation:** 3

**Summary:**

This paper introduces the IDUM method for predicting uplift in an out-of-distribution setting. It utilizes a Gumbel Softmax-based feature selection mechanism to identify a relevant subset of features, followed by invariant property learning. Additionally, the balancing discrepancy component mitigates selection bias, improving model robustness. Through empirical evaluations, the authors claim that these three components collectively enhance uplift prediction in online marketing.

**Claims And Evidence:**

The authors evaluate their proposed methods against baseline models using AUUC, QINI, and KENDALL metrics on two real-world datasets.

**Essential References Not Discussed:**

NA

**Experimental Designs Or Analyses:**

Looks good to me.

**Methods And Evaluation Criteria:**

Evaluation criteria:
It would be helpful if the authors could include the definitions of AUUC and Qini coefficients used in this work. Typically, Qini coefficients are defined as the difference between AUUC and the area under the curve of a random model, which differs from the authors' description that it "scales the responses in the control group."

**Other Comments Or Suggestions:**

None.

**Other Strengths And Weaknesses:**

Strengths:
- Provides a comprehensive literature review of related works, including uplift modeling and its applications in online marketing.
- Conducts analysis on real-world datasets to compare the proposed model with baseline methods.

Weaknesses:
- The work appears to lack novelty, as all three components are derived from previous research. The theoretical framework is entirely based on Yang et al. (2024) [Invariant Learning via Probability of Sufficient and Necessary Causes], and the model optimization objective follows the same paper, with the addition of a distribution discrepancy regularizer.
- Uplift modeling in online marketing provides better customer targeting strategies. The authors could include an online experiment to justify the proposed methods' effectiveness.

**Questions For Authors:**

1. Could the authors clarify the settings of the Lazada dataset? Based on my understanding, the treatments in this dataset correspond to different voucher distribution strategies. What is the response variable (i.e., the y values) used in the model?
2. The proposed IDUM method exhibits an unusually large variance in AUUC on both the ID and OOD Lazada datasets, with a standard error of 0.02, whereas other methods have standard errors around 0.002–0.003. Could the authors investigate and explain the reason for this magnitude difference?

**Relation To Broader Scientific Literature:**

This paper integrates three previous works for application in uplift modeling. The invariant property learning approach is based on Yang et al. (2024) [Invariant Learning via Probability of Sufficient and Necessary Causes], with all definitions, lemmas, properties, and theorems directly drawn from this work. The feature selection mechanism is adapted from Jang et al. (2016) [Categorical Reparameterization with Gumbel-Softmax], while the distribution discrepancy regularizer is based on Shalit et al. (2017) [Estimating Individual Treatment Effect: Generalization Bounds and Algorithms].

**Theoretical Claims:**

Looks good to me.

---

> ### Author Rebuttal · Authors · 2025-03-31
>
> **Dear Reviewer xxdc**,
>
> Thank you for taking the time to review our work, we sincerely appreciate your insightful comments, which have helped improve our paper. Below, we provide point-by-point responses to each concern raised.
>
> **Methods And Evaluation Criteria:**
>
> We will follow your advice to include the the detailed definitions of AUUC and Qini coefficients and revise the presentation in our final version.
>
>
> **Weaknesses:**
>
> **1. Three components are derived from previous research**
>
> Our main contribution is to solve the OOD problem in uplift modeling by introducing the PNS risk, which has been not fully investigated by previous works.
>
> In our work, we build upon existing methods to construct our model architecture. However, due to differences in application scenarios, additional components are required to address the problem effectively. For OOD uplift modeling, we account for the **in-distribution shift** between treatment and control groups by employing a discrepancy regularizer—a common solution that is not our key contribution. To address the PNS risk, we introduce a novel intervention and prediction procedure for balanced embedding learning, which distinguishes our approach from prior work. However, we observed that the **computational cost** of PNS-based feature optimization was prohibitively high. To mitigate this, we adopted a Gumbel-Softmax-based feature selection mechanism.
>
> The theoretical foundations of our work are primarily based on Pearl's causal framework, with additional insights adapted from Yang et al. (2024) to enhance rigor. However, our approach is not a direct application of their theorems, but rather a tailored extension for uplift modeling. First, our focus differs fundamentally from Yang et al. (2024): while they address image classification, we tackle uplift estimation with dual prediction heads. These domains differ critically in objectives and assumptions—for instance, uplift modeling requires counterfactual reasoning absent in standard classification. This divergence necessitates theoretical adaptations. Second, we explicitly model the interaction between prediction heads and balanced embeddings within our theoretical framework, which is uniquely designed for our problem setting. Finally, all prior works have been properly cited to maintain academic norms and contextualize our contributions.
>
> **2. Online experiments.**
>
> Following your suggestions, we conducted an online experiment to evaluate our proposed IDUM against the baseline CFRNet. Due to the limited time of rebuttal and the cost of algorithm deployment, we only keep it for three days.
>
> **Experimental Setup:**
>
> In-Distribution (ID) Test: We selected two user groups with similar distributions to the training data.
>
> Out-of-Distribution (OOD) Test: We used two user groups with distinct distributions to assess generalization.
>
> **Evaluation Protocol:**
>
> Due to the constraints of online inference, we could not obtain the true responses for users unaffected by the deployed algorithm. Instead, we compared the watch time improvement across experimental groups as the key metric.
>
> - **Watch time evaluation**
> | Method  | ID    | OOD    |
> |---------|-------|--------|
> | CFRNet  | 0     | 0      |
> | IDUM    | 0.012%| 0.028% |
>
> - **Cost evaluation**
> | Method  | ID    | OOD    |
> |---------|-------|--------|
> | CFRNet  | 0     | 0      |
> | IDUM    | -1.21%| -1.75% |
>
>
> We report the percentage of watch time improvement relative to CFRNet's performance (baseline, set at 0). And also, we evaluate the cost of two methods, for that we assign less incentives (-1.21% and -1.75%) to the user group and get competitive performance (0.012% and 0.028%).
>
> The results demonstrate that our proposed IDUM achieves further gains in the OOD testing environment.
>
>
>
>
> **Questions**
>
>
> **1. clarify the settings of the Lazada dataset**
>
> The Lazada dataset is a public dataset for uplift modeling, which can be found at: https://drive.google.com/file/d/19iSXsbRXJWvuSFHdcLb0Vi9JCP9Fu41s/view
>
> In the dataset description, there are 86 fields in total, of which 83 are features(f0-f82). We use the column of is_treat as treatment $t$, and the column label as the response $y$.
> Usually, for uplift modeling, $t$ represents whether the user is assigned the discount or coupon, $y$ represents whether the user is converted.
>
> **2. The variance**
>
> We apologize for the error in reporting the variance. As you pointed out, the variance was incorrectly magnified by a factor of 10. We will correct this in the final version of our paper. Additionally, we plan to release the code, dataset, and training logs upon obtaining permission from our collaborating company, and the variance can be identified in the training logs. Thank you for your careful attention to this issue.
>
> If you have any further questions to discuss, we are willing to reply as soon as possible.

---

### Decision · Program_Chairs · 2025-05-01

**Decision:**

Accept (spotlight poster)

**Comment:**

This paper addresses a practical problem in uplift modeling for online marketing, aiming to tackle out-of-distribution generalization challenges and selection bias by eliminating spurious correlations. Its emphasis on causal invariance and the decomposition of feature sufficiency and necessity helps bridge important gaps in real-world deployment scenarios. While the core concepts are not entirely novel, their application to these scenarios is valuable. Most of the reviewers' concerns were satisfactorily addressed in the rebuttal. However, the authors are encouraged to incorporate the reviewers' suggestions into the final version.